# Annual oil palm plantation maps in Malaysia and Indonesia from 2001 to 2016

Yidi Xu[1], Le Yu[1,2*], Wei Li[1], Philippe Ciais[3], Yuqi Cheng[1], Peng Gong[1,2]

[1]Ministry of Education Key Laboratory for Earth System Modeling, Department of Earth System Science, Tsinghua University, Beijing, 100084, China
[2]Joint Center for Global Change Studies, Beijing 100875, China
[3]Laboratoire des Sciences du Climat et de l'Environnement, LSCE/IPSL, CEA-CNRS-UVSQ, Universite Paris-Saclay, Gif-sur-Yvette 91191, France

*Correspondence to*: Le Yu (leyu@tsinghua.edu.cn)

**Abstract.** Increasing global demand of vegetable oils and biofuels results in significant oil palm expansion in Southeast Asia, predominately in Malaysia and Indonesia. The land conversion to oil palm plantations poses risks to deforestation (50% of the oil palm was taken from forest during 1990-2005, (Koh and Wilcove, 2008)), loss of biodiversity, and greenhouse gas emission over the past decades. Quantifying the consequences of oil palm expansion requires fine scale and frequently updated datasets of land cover dynamics. Previous studies focused on total changes for a multi-year interval without identifying the exact time of conversion, causing uncertainty in the timing of carbon emission estimates from land cover change. Using Advanced Land Observing Satellite (ALOS) Phased Array Type L-band Synthetic Aperture Radar (PALSAR), ALOS-2 PALSAR-2 and Moderate Resolution Imaging Spectroradiometer (MODIS) datasets, we produced an Annual Oil Palm Area Dataset (AOPD) at 100-meter resolution in Malaysia and Indonesia from 2001 to 2016. We first mapped the oil palm extent using PALSAR/PALSAR-2 data for 2007-2010 and 2015-2016 and then applied a disturbance and recovery algorithm (BFAST) to detect land cover change time-points using MODIS data during the years without PALSAR data (2011-2014 and 2001-2006). The new oil palm land cover maps are assessed to have an accuracy of 86.61% in the mapping step (2007-2010 and 2015-2016). During the intervening years when MODIS data are used, 75.74% of the change detected time matched the timing of actual conversion using Google Earth and Landsat images. The AOPD dataset revealed spatiotemporal oil palm dynamics every year and shows that plantations expanded from 2.59 to 6.39 M ha and from 3.00 to 12.66 M ha in Malaysia and Indonesia, respectively (i.e., a net increase of 146.60% and 322.46%) between 2001 and 2016. The higher trends from our dataset are consistent with those from the national inventories with limited annual average difference in Malaysia (0.2 M ha) and Indonesia (-0.17 M ha). We highlight the capability of combining multiple resolution radar and optical satellite datasets in annual plantation mapping at large extent using image classification and statistical boundary-based change detection to achieve long time-series. The consistent characterization of oil palm dynamics can be further used in downstream applications. The annual oil palm plantation maps from 2001 to 2016 at 100 m resolution is published in the Tagged Image File Format with georeferencing information (GeoTIFF) at https://doi.org/10.5281/zenodo.3467071.

# 1 Introduction

The global demand for vegetable oil and its derivative products calls for an increase in palm oil production leading to oil palm expansion and intensification in Southeast Asia (Sayer et al., 2012). According to the Food and Agriculture Organization (FAO), Malaysia and Indonesia account for 81.90% of the global oil palm fruit production in 2017, an increase by 179.72% from 2000 to 2017 (see http://faostat.fao.org) that is projected to continue in the future (Murphy, 2014). The boom of oil palm industries caused and also raised the deforestation risks (Austin et al., 2018; Vijay et al., 2018). In Malaysia and Indonesia, more than 50% of the oil palm plantation was converted from forest during 1990-2005 (Koh and Wilcove, 2008) and industrial plantation dominated by oil palm (72.5% of all plantations) caused a ~60% decrease of peatland forest from 2007 to 2015 (Miettinen et al., 2016). A series of consequences include but not limited in biodiversity decline (Fitzherbert et al., 2008), peatland loss (Koh et al., 2011) and carbon emission (Guillaume et al., 2018).

Quantifying the spatiotemporal details of oil palm expansion is important to understand the deforestation process and its impacts on ecosystems services and promote progress in environmental governance and policy decisions (Gibbs et al., 2010; Koh and Wilcove, 2008). However, annual information on the expansion of oil palm plantations is poorly documented in Malaysia and Indonesia. The statistical records (e.g., FAO, United States Department of Agriculture (USDA)) give neither the detailed spatial distribution nor the young oil palm trees and small-holder plantations. Many efforts have been made to characterize the oil palm extent (Cheng et al., 2018; Gaveau et al., 2016; Miettinen et al., 2017). For example, the Roundtable on Sustainable Palm Oil (RSPO), whose members manage 1/3 of the world's oil palm, provided spatial information on oil palm distribution in Malaysia and Indonesia (Gunarso, 2013). The continuous mapping of oil palm on peatland in 1990, 2000, 2007 and 2010 described the dynamic change of oil palm on peat during the past 30 years (Miettinen et al., 2012). But these maps are given for a certain year or several time phases without capturing the exact time of oil palm changes. Dynamic global vegetation models use gross land-use change and thus require high-resolution grid-cell-based annual oil palm conversion maps rather than country-level inventories and bi-decadal land cover maps (Yue et al., 2018a; Yue et al., 2018b). Lack of continuous change information may cause wrong interpretation of land cover change time and significant bias in global carbon dynamic studies (Zhao and Liu, 2014; Zhao et al., 2009). As a result, oil palm plantation maps at high temporal and spatial resolutions in Malaysia and Indonesia are urgently needed.

Remote sensing has been used in oil palm monitoring since 1990s. Progress has been made in oil palm mapping and change detection, including 1) data sources from optical satellite earth observations (Lee et al., 2016; Srestasathiern and Rakwatin, 2014) to microwave datasets such as Phased Array Type L-band Synthetic Aperture Radar (PALSAR) (Cheng et al., 2018; Dong et al., 2015), 2) spatiotemporal resolutions from regional to national scale (Miettinen et al., 2017) and from single to multi-decadal mapping (Gaveau et al., 2016; Miettinen et al., 2016), 3) interpretation methods from manual to semi- and fully automatic identification (Baklanov et al., 2018; Cheng et al., 2019; Li et al., 2017a; Mubin et al., 2019; Ordway et al., 2019), 4) products going from oil palm land cover maps to more detailed datasets on plantation structure, e.g. tree counting (Li et al., 2019; Cheang et al., 2017) age and yield estimation (Balasundram et al., 2013; Tan et al., 2013) and etc. A few studies also

focused on the continuous oil palm change detection (Carlson et al., 2013; Gaveau et al., 2016; Vijay et al., 2018). These studies adopted visual or semi-automatic interpretation for oil palm plantation, which is labor-extensive and not appropriate for long-term annual oil palm plantation monitoring. Automatic identification can overcome this difficulty by using classification algorithms based on Landsat and PALSAR/PALSAR-2 data, which were successfully applied to produce the 2015 land cover map of insular Southeast Asia with discrimination of oil palm plantation (Miettinen et al., 2017). So far, however, the annual dynamics of oil palm plantations (expansion and shrinkage) remains unquantified for Malaysia and Indonesia.

The annual oil palm mapping in tropical areas such as insular South-East Asia is a challenge due to the persistent cloudy conditions (Gong et al., 2013; Yu et al., 2013). Multi-temporal optical images can help reduce cloud effects (Yu et al., 2013) but it is still difficult to obtain effective optical observations in Malaysia and Indonesia (51.88% of the region is without annual Landsat images, Figure S1). Microwave remote sensing is not affected by clouds, and is considered to be the most efficient source in separating forested vegetation and oil palms (Ibharim et al., 2015; Teng et al., 2015). The long-time span of 25 m resolution PALSAR/PALSAR-2 data provides opportunities for mapping oil palm at high spatiotemporal resolutions. Recently the PALSAR/PALSAR-2 data have been successfully used in charactering oil palm change for the whole Malaysia for six years using PALSAR (2007-2010) and PALSAR-2 (2015-2016) (Cheng et al., 2019). However, the gap years (2011-2014) between PALSAR and PALSAR-2 hampered continuous tracking of oil palm plantation dynamics. One potential way to achieve annual mapping is to use optical earth observation data e.g., Landsat images for the PALSAR gap period (Chen et al., 2018; Shen et al., 2019). However, this requires abundant Landsat images (>4) (Xu et al., 2018a) that are not available in the humid tropical regions and may cause "false changes" and "inter-annual inconsistency" (Broich et al., 2011). Recently, a super-resolution mapping method (Li et al., 2017b; Qin et al., 2017; Xu et al., 2017) was used to reconstruct missing forest cover change during 2011–2014 (Zhang et al., 2019) by fusing the PALSAR/PALSAR-2 and the MODIS normalized difference vegetation index (NDVI) with dense temporal resolution and phenological information. However, it is difficult to separate oil palm and natural forest with similar NDVI variation using such classification-based fusion. A new approach based on change detection in a given period using time-series observations (i.e., MODIS NDVI, GIMMS NDVI) was successfully applied to fill the data-missing years in developing a nominal 30 m annual China land use and land cover dataset (Xu et al., in review). This approach takes advantage of dense observations by detecting break points in a time-series using change detection algorithms, combined with the pre-knowledge from the mapped years and thus reduces the inter-annual inconsistency.

The objectives of this study are (i) to develop a robust and consistent approach capable of detecting annual oil palm changes in Southeast Asia using multiple remote sensing datasets based on image classification and breakpoint detection, (ii) to produce a nominal 100 m annual oil palm plantation dataset (AOPD) in Malaysia and Indonesia from 2001 to 2016, and (iii) to quantify the spatial and temporal patterns of oil-palm change dynamics since 2001. Specifically, we developed the annual oil palm plantation dataset in Malaysia and Indonesia by using a two-stage method. The first step is random forest-based image classification using PALSAR during 2007-2010 and PALSAR-2 data during 2015-2016 (the periods with PALSAR/PALSAR-2 data available). Combined with the oil palm maps produced in the first step during the years with PALSAR coverage, MODIS

NDVI was used in a change detection algorithm called Breaks for Additive Seasonal and Trend (BFAST) (Verbesselt et al., 2010a), to fill the data-gap years (2011-2014) outside the PALSAR years and extend the oil palm land cover mapping period back to 2001. Oil palm in this study refers to both young and mature oil palm trees from industrial plantation and smallholders with the minimum size of 1 ha (oil palm smallholders is defined as 50 hectares or less of cultivated land producing palm oil controlled by smallholder farmers (the definition used by the RSPO) with an average of 2 ha (Bank, 2010)).

## 2 Datasets and method

### 2.1 Study area

Insular South-East Asia was originally occupied by evergreen moist tropical forest, which is one of the most biologically diverse terrestrial ecosystem on Earth. The natural environment, with humid tropical climates and low-lying topography, is suitable for the oil palm (Elaeis guineensis) (Fitzherbert et al., 2008). Since 1911 when the first commercially oil palm plantation in Southeast Asia settled in Sumatra, oil palm plantation expanded rapidly in Sumatra and peninsular Malaysia and then spread to Sarawak and Sabah in Malaysia and Kalimantan in Indonesia (Corley and Tinker, 2008). Industrial oil palm plantations spurred the economic sectors in Southeast Asian countries but also raised concerns on the negative social and environmental impacts (Obidzinski et al., 2012; Sayer et al., 2012). Recently, oil palm plantations expansion became one of the dominant drivers of deforestation in Malaysia and Indonesia (Austin et al., 2018; Gaveau et al., 2016). Thus, we chose as a study area the whole Malaysia, Sumatra and Kalimantan in Indonesia, encompassing 96% of the total oil palm production in Indonesia (Petrenko et al., 2016). Oil palm plantations in these two countries account for 67.51% of world's total oil palm plantation area (FAOSTAT, 2017), and dramatic land cover conversion happened in this region due to human induced modifications.

### 2.2 Overview of the AOPD producing

The development of AOPD includes two major stages: 1) oil palm mapping using PALSAR/PALSAR-2 data (Section 2.3) and 2) change-detection based oil palm updating using MODIS NDVI during the gap years in operation between ALOS and ALOS-2 (Section 2.4). The first stage aimed at producing the oil palm maps for 2007, 2008, 2009, 2010 using PALSAR and 2015, 2016 using PALSAR-2 datasets. The detailed procedures include the pre-process of the original PALSAR/PALSAR-2 data, training sample collection and image classification and final production of oil palm maps for the target years after post-processing using ancillary datasets. In the second stage, we combined oil palm maps produced in the first stage with MODIS NDVI data. Time series of MODIS NDVI data and change maps were prepared in the data preparation step, followed by the breakpoint test using change-detection algorithm, BFAST to detect the change year (change from other land cover types to oil palm and the reverse) in the PALSAR/PALSAR-2 data missing period. After the post-processing, we derived the oil palm maps in these gap years and traced the oil palm distribution back to 2001. Combining the results from the two stages we

obtained the annual oil palm plantation maps from 2001 to 2016 at 100 m spatial resolution, forming the AOPD dataset. The whole workflow is shown in Figure 1.

## 2.3 Oil palm mapping using PALSAR/PALSAR-2 data

### 2.3.1 PALSAR/PALSAR-2 product and data preparation

We used multi-source remote sensing images to fully cover the whole study period including ALOS PALSAR, ALOS-2

PALSAR-2 and MODIS NDVI. The Landsat archives were not used because of the low data availability in this region caused by frequent thick cloud cover (Figure S1).

Japan Aerospace Exploration Agency (JAXA) provided the 25 m resolution global PALSAR/PALSAR-2 mosaic by mosaicking SAR images of backscattering coefficient (http://www.eorc.jaxa.jp/ALOS/en/palsar_fnf/data/index.htm). Although the product was compiled at an annual frequency, one product a year is sufficient to identify the oil palm changes

since oil palm is a perennial crop without significant phenological variations in the tropics. To cover the whole study area, 15 patches of 5°×5° PALSAR/PALSAR-2 grids for six years (2007, 2008, 2009, 2010 from PALSAR; 2015, 2016 from PALSAR-2) were used. Since ALOS satellite stopped working in 2011, no data was available between 2011 and 2014 until the operation of ALOS-2. The product contains data of HH (i.e. horizontal transmit and horizontal receive) and HV (i.e. horizontal transmit and vertical receive) digital numbers ($DN$) acquired by PALSAR/PALSAR-2 in Fine Beam Dual (FBD) mode with ortho-

rectification and topographic correction. For PALSAR/PALSAR-2, HH and HV DN values were converted to normalized backscattering coefficients (unit: decibel (dB)) using the following Eq. (1) formula (Rosenqvist et al., 2007):

$$\sigma^0(\text{dB}) = 10 \times log_{10}DN^2 + CF , \qquad (1)$$

where $CF$ is a calibration factor ($-83.0$ dB) in PALSAR/PALSAR-2 data (Shimada et al., 2009). Two additional layers, $Difference$ and $Ratio$, were produced by calculating the ratio and difference from $HH$ and $HV$ $DN$ of decibels as followings

Eq. (2) and Eq. (3):

$$Difference = HH - HV , \qquad (2)$$

$$Ratio = HH/HV , \qquad (3)$$

Although the ALOS PALSAR and ALOS-2 PALSAR-2 have different satellite microwave sensor properties (e.g., frequency, off-nadir angle), the backscatter signals are relatively stable for the given period (2007–2010 and 2015–2016) as seen by

comparing the distribution of backscattering values (HH and HV) of 250000 randomly generated pixels (using ArcGIS 10.3) in the study area between different years (see Figure S2). The similar findings for the stability of PALSAR/PALSAR-2 data was also given in previous studies (Cheng et al., 2019; Qin et al., 2017). Meanwhile, the HH and HV values for oil palm and forest is also shown in Figure S3 and indicate the separability between the two land cover types for both PALSAR/PALSAR-2 data. Therefore, the consistency between ALOS PALSAR and ALOS-2 PALSAR-2 allows tracking the oil palm changes in

the study period. One problem of using PALSAR/PALSAR-2 data, however, is the "salt and pepper" noise (Zhang et al., 2019), which may cause misclassification and false changes in the subsequent process. Previous studies showed that the resampling method reached higher accuracy and better visual results in oil palm mapping compared to the commonly used filter method (Cheng et al., 2018). The identification and area estimation of oil palm plantations have also been proven to perform better at 100 m resolution (Cheng et al., 2018). Therefore, we resampled the original 25 m PALSAR/PALSAR-2 images to 100 m resolution for every year to reduce "salt and pepper" noise.

### 2.3.2 Training sample collection and image classification

In this study, a multi-year training sample set (2007-2010, 2015 and 2016) was used to map the oil palm extent in Indonesia and Malaysia from 2007 to 2016. We used the training sample set for Malaysia from our previous study (Cheng et al., 2017) and interpreted the training datasets for Indonesia using the same interpretation method. The sample collection was mainly based on the high-resolution (<1m) images from Google Earth with the assistance of PALSAR/PALSAR-2 images. We first visually interpreted the samples in 2015 and then manually checked the land cover types forwards and backwards if change happened. Here we used 636 and 748 polygonal regions of interest (ROIs) (4953-5660 and 7804-8147 pixels) for Malaysia and Indonesia as the training inputs instead of point sample-based training since it achieved better results in regular plantations. Four land cover types in this training sample set were included: oil palm (mature and young oil palm—identified by the canopy shape using very high-resolution images from Google Earth), water, other vegetation (forest, shrubland and other plantations such as rubber), and others (impervious, cropland and bare land). Mixed land cover types were found in "other vegetation" and "others" because it is difficult to further separate these types within the categories. The detailed distribution of training data is presented in Table 1. Other vegetation types consist of ~52.9% of the total sample, secondly ranked the oil palm samples (26.7%), while "others" and water types only account for ~20.4% of the total training samples, which is consistent with the real land cover distribution.

Thereafter, we used a random forest (RF) classifier in the image classification step. The $HH$ and $HV$ digital number of decibels, the derived difference ($HH-HV$) and ratio ($HH/HV$) images were all used as inputs to the RF classifier to derive the original annual oil palm maps for the six years. The MODIS NDVI is not used as input to RF model for classification because of the similarity between tropical forest and oil palm and the coarse resolution which may negate the benefits of our classification based on higher spatial resolution PALSAR data.

### 2.3.3 Post-processing and oil palm map

Post-processing after the initial results is necessary because of the limitation in training set, unavoidable classification errors and the difficulty in describing heterogeneous real land surface. To obtain reliable oil palm dataset, we adopted several steps including mode filtering, terrain filtering, intact forest and mangrove filter in post-process to improve the final oil palm maps in stage 1 for 2007, 2008, 2009, 2010, 2015 and 2016.

Mode filtering is used to filter the very small patches (mainly single pixel) in the initial results since it is more likely to be errors or noise induced by PALSAR/PALSAR-2 data rather than real oil palm plantation. The topographic factor such as slope and elevation will cause the confusion of backscattering signals from satellite sensors, particularly in area with undulating terrain. Therefore, we applied terrain filter to reduce the confusion by topographic factor using the Shuttle Radar Topography Mission (SRTM) 30-m digital elevation model (DEM). The altitude threshold of 1000 m was applied since the oil palm is mainly distributed in lowland (mostly <300 m) and regions higher than 1000 m are not suitable for oil palm cultivation (Austin et al. 2015; Carlson et al. 2013; Corley and Tinker 2008). Subsequently, we used two additional layers, intact forest landscape (IFL) in 2016 from (Potapov et al., 2008) and the Global Mangrove Atlas (GMA, available at: http://geodata.grid.unep.ch/results.php) to filter out non-oil palm areas and reduce the misclassification. The intact forest map denotes natural forest ecosystem without human caused disturbances where oil palm plantation is not supposed to be cultivated. The mangrove swamp area is subsequently flooded by sea water, which is not suitable for oil palm cultivation due to the significant negative impact on the fresh fruit bunch and oil production (Henry and Wan, 2012).

Another problem when developing oil palm maps is the replantation of oil palm trees. Oil palm has a long-life cycle of 25 to 30 years. After that, the trees will be cleared and replaced because of a decrease in palm oil yield (Röll et al., 2015). However, from the satellite observations, the land cover type is bare land at the time of oil palm logging whereas the land use property remains unchanged as oil palm plantation backwards and forwards. Given the limitation of satellite observation, we provided two versions of our oil palm datasets. The first version is the oil palm datasets after the post-processing mentioned above. Here replantation is not considered, and this version includes conversion from other land cover types to oil palm (oil palm expansion) as well as the opposite one (oil palm shrinkage). In the second version, we assumed that oil palm expansion is a unidirectional activity due to the growing demand of palm oil. The time-series filtering was conducted by using the 2007 oil palm extent to filter all pixels classified as "non-oil palm" in the subsequent years. As a result, this version of the oil palm plantation dataset has continuously expanding areas from 2007 to 2016. The second version includes the impact of oil palm replantation and the thriving oil palm industry in South-East Asian countries but ignored any possible decrease of oil palm (e.g. abandonment, conversion to cropland) in some areas.

**2.4 Change-detection based oil palm updating using MODIS NDVI**

**2.4.1 MODIS NDVI time-series and data preparation**

MODIS NDVI is an important index of vegetation conditions and has been widely used in vegetation and land cover change studies (Clark et al., 2010; Ding et al., 2016; Estel et al., 2015). NDVI in the recent updated MODIS vegetation index data (MOD13Q1) collection 6 from 2000-2007 and from 2010-2015 (downloaded from https://lpdaac.usgs.gov/) was used to fill the gap years (2000-2006 and 2011-2014) of PALSAR/PALSAR-2 datasets using change detection algorithms. The MOD13Q1 product has a spatial resolution of 250 m and is composited every 16 days. In total, 6 MODIS tiles with 23 scenes per year (181 and 138 scenes for the two study periods, 2000-2007, P1 and 2010-2015, P2) were required to cover the study

area (h27v08, h27v09, h28v08, h28v09, h29v08 and h29v09). All the MODIS images were projected from its original sinusoidal projection to a geographic grid with a WGS 1984 spheroid and resized to 100 m to match the resolution of the oil

palm maps using the nearest neighbor resampling approach. The pixel quality and reliability layers in the MOD13Q1 product were used to further exclude the poor-quality pixels. During the whole study period, 53.64% of the observations have good quality while 46.36% were interpolated using spline interpolation. For those pixels with less than 30 good-quality observations (4.79% in P1 and 9.64% in P2 of the total change area), we didn't apply the BFAST algorithm. For the remaining area, 61.67% (P1) and 58.24% (P2) of pixels had 12 (~50% in 23) good-quality observations annually.

A change map for the microwave data gap period between PALSAR and PALSAR-2 (2011-2014) was extracted using the change pixels in 2010 and 2015 oil palm maps with spatial locations and "from-to" types. Here, we assumed the change from classification was reliable because of the high resolution of PALSAR data. We then sought the exact change year within the intervals in the next step (Section 2.4.2) using temporal NDVI files extracted from each change pixel. Frequent changes such as two or three shifts during the gap years were assumed to be of low probability and thus not considered in this study. For the

period during 2001-2006 without PALSAR/PALSAR-2 data and oil palm distribution in 2000, we assumed a unidirectional expansion of oil palm and the oil palm extent in 2007 was used as the potential change regions in the next step. In total, we derived two versions of change maps (one with bi-directional change and the other with only unidirectional oil palm expansion) for the two periods.

## 2.4.2 Breakpoint test using change-detection algorithm, BFAST

Change detection analysis was conducted in the change pixels derived from the last step to identify the exact change time within the two periods (2011-2014 and 2001-2006) based on the time-series MODIS NDVI from 2010 to 2015 and 2000 to 2007, respectively. Here we aimed to capture an abrupt NDVI changes (breakpoints) in the two given periods, which is assumed to be caused by the conversion of the original land cover type to the oil palm cultivation. Many change detection algorithms and their derivatives have been developed in recent years to detect subtle or abrupt changes in a dense time-series

satellite profiles (Broich et al., 2011; Kennedy et al., 2010; Verbesselt et al., 2010b). Most of these algorithms were applied in forest change monitoring and all reach high consistency in detecting significant change (Cohen et al., 2017). A recent algorithm, Bayesian Estimator of Abrupt change, Seasonal change, and Trend (BEAST), aggregating the competing models than the conventional single-best-model, performed well in capturing multiple and subtle phenological changes (Zhao et al., 2019b). Here we used BFAST to capture the oil palm conversion time within the two study periods (2011-2014 and 2001-2006).

BFAST has been successfully applied in monitoring forest disturbance and regrowth and has proved robust with different sensors (DeVries et al., 2015; Verbesselt et al., 2012). Based on the structural change methods, the BFAST algorithm is able to find the structural breakpoint between different segments in the observation time series (DeVries et al., 2015), and thus can be used to detect the time and number of abrupt or gradual changes as well as to characterize the magnitude and direction. The BFAST method decomposes the time series into trend, seasonality, and residuals sections (Verbesselt et al., 2010b). The model

can be expressed as Eq. (4):

$$Y_t = T_t + S_t + e_t(t = 1, \dots, n),  \tag{4}$$

where $Y_t$ is the observed value at time $t$, $T_t$ is the trend section, $S_t$ is the seasonal section and $e_t$ is the noise section.

An ordinary least square residuals-based moving sum test (Zeileis, 2005) was used to test whether breakpoints occurred in the trend or seasonal components. Then, test was conducted to determine the number and optimal position of the breaks using Bayesian Information Criteria and the minimum of the residual sum of squares. The trend and seasonal coefficients were then computed using robust regression. A harmonic seasonality model (with three harmonic terms) was used to describe the seasonality of the satellite data (Eq. 6) (Verbesselt et al., 2010b). For each piecewise linear $(T_t)$ from $t_i^*$ to $t_{i+1}^*$ where $t_1^*, \dots, t_p^*$ is the assumed break points which defines the $p+1$ segment, $T_t$ can be expressed as follows:

$$T_t = \alpha_i + \beta_i t \; (i = 1, \dots, p),  \tag{5}$$

where $i$ is the index of the breaks, $i=1, \dots, q$. $\alpha_i$ and $\beta_i$ are the intercept and slope of the fitted piecewise linear model.

For the $t_1^\#, \dots, t_m^\#$ seasonal break points, $S_t$ is the harmonic model for $t_j^\#$ to $t_{j+1}^\#$:

$$S_t = \sum_{k=1}^{K} \alpha_{j,k} \sin(\frac{2\pi k t}{f} + \delta_{j,k}) \; (j = 1, \dots, q)  \tag{6}$$

where, $j = 1, \dots q$. $k$ is the number of harmonic terms in the periodic model (default value $= 3$); $\alpha_{j,k}$ is the amplitude; $f$ is the frequency; $\delta_{j,k}$ is the time phase. For the MODIS NDVI used in this study, the $f$ value was 23 (i.e. 23 observations of MODIS observations per year) (Verbesselt et al., 2010b). Here, the maximum number of breaks was artificially set to 1 because of the assumption of one time change for each period based on prior knowledge from the oil palm maps. Figure 2(a) shows two examples of the breakpoint detection of the MODIS NDVI using BFAST algorithm. In the first example, no obvious break detected in the coarse resolution time-series, whereas significant change was captured in the trend section after time-series decomposition in the second example (Figure 2(b)). More details of the BFAST algorithm are referenced in (Verbesselt et al., 2010b; Verbesselt et al., 2012). To evaluate the validity of using coarse MODIS time series in oil palm change detection, we visually interpreted an extra 100 change points based on the PALSAR images from 2007 to 2010 and applied the BFAST algorithms using the MODIS NDVI. An example of the comparison between the BFAST based change results and visual interpretation from PALSAR images was shown in Figure S4. The break time detected from MODIS NDVI showed the similar conversion year compared with the microwave satellite images (a total of 86% agreement with 62% matched the same change year and 24% within a 1-year interval). Moreover, we did a test using the subsequent NDVI fragments to replace the original NDVI fragments after the break detected time and compared the break results to show the robustness of the algorithm considering the effect of oil palm plantation stand age (Figure S5).

### 2.4.3 Annual oil palm results updating

The previous steps generated annual oil palm maps for six years (Section 2.3) and the oil palm change time in the missing periods (2011-2014 and 2001-2016) (Section 2.4.1 and 2.4.2). In the final step, all these data were combined to update the continuous oil palm dataset from 2001 to 2016 following Xu et al (under review).

For the gap period from 2011 to 2014, the oil palm updating was based on the "from-to" land cover types ($L_1$ and $L_2$) of the start ($t_1$) and the end years ($t_2$) with the detected change time ($t_i$). Then $L_2$ was allocated between $t_i$ and $t_2$ while $L_1$ was assigned before $t_i$ ($t_1$ to $t_i$). For example, if a pixel was forest in 2010 and oil palm in 2015 with a change year of 2013, then the land cover type would be forest during 2010-2013 and oil palm during 2014-2015 following the updating process. The rest of the area without oil palm changes remained unchanged from 2010 to 2015 (assigned $L_1$). For the gap period during 2001-2006, the oil palm map in 2007 from PALSAR data was used as the potential change area (as mentioned in 2.4.2) without "from-to" types. So, the land cover type between 2001 and change time ($t_i$) was classified as non-oil palm, and oil palm was assigned to the period after $t_i$ ($t_i$ to $t_2$). Thereafter, the oil palm maps between 2001 to 2016 were updated. Quality maps (Figure S6 and S7) were also generated to indicate the availability of valid NDVI values (i.e., not under cloud cover), the spatial resolution of the dataset used and the consistency of change time detection from different breakpoint test approaches in BFAST algorithms (the ordinary least squares residuals-based MOving SUM test (OLS-MOSUM), the supremum of a set of Lagrange multiplier statistics (SupLM) and Bayesian information criterion test (BIC), (Zeileis, 2005)). The annual oil palm updating process was applied in both the bi-directional and unidirectional versions. And two versions of the oil palm datasets (AOPD-bi and AOPD-uni) were developed.

### 2.5 Evaluation

Our product of annual oil palm maps, AOPD, was evaluated in three aspects: 1) independent annual oil palm sample set for Malaysia (2007, 2008, 2009, 2010, 2015 and 2016) and Indonesia (2010-2016) to evaluate the annual mapping results for the classified maps using PALSAR/PALSAR-2 data and gap years using change detection method, 2) a change sample set aimed at assessing the accuracy of detected change years and 3) comparison with statistical inventories (e.g., FAO, USDA, Malaysian Palm Oil Board (MPOB) (2011-2016), Badan Pusat Statistik (BPS-Statistics Indonesia) (2011-2015)), the existing oil palm maps from Gaveau et al. (2016) and the Landsat based deforestation maps (Hansen et al., 2013). FAO and USDA agricultural statistical data provided the harvested area of oil palm using data collected by official and unofficial outlets. MPOB is a government agency providing oil palm planted area in Malaysia based on the data reported by state agencies, institutions, private estates and independent smallholders. BPS-Statistics Indonesia, a non-ministry government agency, provided statistical data for public including oil palm planted area compiled from Quarterly (SKB17-Oil Palm) and Annually (SKB17-Annual) Plantation Estate Survey, custom documents from Directorate General of Customs and secondary data from Directorate General of Estate Crops.

Two sets of annual oil palm samples set were used to validate the mapping results in Malaysia and Indonesia according to the sampling protocol of Gong et al. (2013). The independent annual sample set in Malaysia was from the previous studies (Cheng et al., 2019; Cheng et al., 2017). All pixel-based samples were randomly produced in equal-area hexagonal grid (95.98 km$^2$ for each grid cell), therefore the distribution of the samples among different land cover types has minimum bias with the real land cover composition. All the testing samples were manually checked using high-quality Google Earth (<1 m) at the first round and then double checked by the time-series PALSAR images (25 m) since it is easy to identify the crown of palm trees in the high-resolution datasets and recognize the regular oil palm plantations in the microwave satellite datasets. Once the start and the end of the period is determined of oil palm based on Google Earth images or PALSAR data, the middle years can be checked by the stable spectral/backscatter coefficient information in the continuous PALSAR images. The annual sample set contains ~3000 samples with four land cover types (~16% were oil palm samples) and it covers the whole Malaysia (see the green points in Figure 3, only oil palm samples presented). The second annual Indonesia sample set was developed following the protocol of (Cheng et al., 2017). This sample set contains 7663 samples in total (601 were oil palms and the rest were non-oil palm types) during 2010 to 2016 (see the blue points in Figure 3). The details of the number and spatial distribution of validation samples is presented in Figure 3 and Table 2. More information on the randomized sampling method could be referred to (Cheng et al., 2019; Cheng et al., 2017).

The change sample set was developed to evaluate the detected change year by the breakpoint detection analysis. Time lapses of high-resolution imagery from Google Earth covering the change period were used to check the change time detected by the BFAST algorithm. We randomly selected 5000 points (implemented with ArcGIS 10.3 software) in the change area but there were only limited samples (370, 25.07% of the total 1476 oil palm samples) with continuous high-resolution images from Google Earth and cloud-free Landsat time series. We compared our detected change years with the actual oil palm conversion time for these test samples. A confidence interval of ±1 years was used considering uncertainty in visual interpretation of the change time (Dara et al., 2018). Detailed distribution of the testing samples can be seen from Figure 3.

# 3 Results

## 3.1 Spatial and temporal characterizes of oil palm expansion

The annual changes of oil palm plantations from 2001 to 2016 are shown in Figure 4. The spatial and temporal dynamics of oil palm changes vary in Malaysia and Indonesia. In the study area, most oil palm plantations are located on lowland areas (elevation <250 m, slope <2.5 degree), and few are distributed in gently undulating hills (elevation >500 m, slope >5 degree) (Figure S8). The newly developed oil palm has similar elevation and slope distribution compared to the 2007 ones (slope: 1.97° in 2007/1.99° in 2016; elevation 228.98 m in 2007/230.10 m in 2016). Specifically, the oil palm plantations are mostly found in the southwest coastal regions in peninsular Malaysia, northeast of Sumatra and coastal regions in Borneo (Figure 4(a)).

Light colors in Figure 4 indicate the oil palm changes (expansion and shrink) at early years while the dark colors are the changes in more recent years. Oil palm plantations expanded rapidly during the study period in peninsular Malaysia and

Sumatra and Borneo. In Indonesia, rapid expansion first occurred in Sumatra and was then surpassed by Kalimantan (Gunarso, 2013; Petrenko et al., 2016). This can also be observed in our maps where more changes happened in earlier years in Sumatra (lighter colors in Figure 4) and later in Kalimantan (darker colors). The decrease in oil palm plantations was also detected (Figure 4(b)), although it is difficult to separate the oil palm replantation after one rotation (i.e. still oil palm in land use) from the permanent oil palm loss (i.e. change to other land use types). Compared to the period before 2007 using change-detection in NDVI data, our data product in the gap period of 2011-2014 would be of better quality since the net changes were constrained by the oil palm maps in 2010 and 2015 derived from PALSAR and PALSAR-2 data, respectively.

Figure 5 displays the annual total area of oil palm in Malaysia and Indonesia from 2001 to 2016 with uncertainty ranges (shaded area with boundary lines) during 2001-2006 and 2011-2014. This uncertainty range is from the change detection step. 9.45% of the total changes from 2010 to 2015 were not captured in the MODIS NDVI using the BFAST algorithm because of the coarse resolution, cloud contamination, the mapping error from the base maps, etc. Assuming that these missing changes all happened from 2010 to 2011, the oil palm area of the gap years should follow the trajectory of the upper boundary line. If all the missing changes happened in the last year of the period, the oil palm area curve would be lower boundary line. Since the distribution of oil palm in 2001 was unknown, large uncertainty may exist before 2007. Here, the uncertainty range during 2001-2006 was determined based on the data availability of MODIS NDVI and consistency of change time detection from the quality maps (Figure S6 and S7). The oil palm area before 2007 follows the upper boundary curve if the same breaks detected in all three structural change methods (OLS-MOSUM, SupLM, BIC) and more than 60% valid NDVI values available in this time period. If all the breaks were counted regardless of the number of valid MODIS NDVI and the consistency of change methods, the oil palm area would be the lower boundary line.

Generally, the net oil palm plantation area shows a monotonous increasing trend from 2001 to 2016 for Malaysia (Figure 5a) and Indonesia (Figure 5b) in both the bi-directional (green lines) and unidirectional (blue lines) versions. During the past 16 years, the net oil palm area across Malaysia increased from ~2.59 M ha (2.05-3.13 M ha) to 6.39 M ha, that is a net increase of 146.60% (103.99-211.71%). Indonesia has much more increase of oil palm area from ~3.00 M ha (1.92-4.07 M ha) to 12.66 M ha (~4-fold). Correspondingly, the increasing trend in oil palm plantation in Indonesia was greater than Malaysia (0.573-0.716 M ha/year compared to 0.217-0.289 M ha/year according to our mapping results), which illustrates the quick expansion of oil palm plantation in Indonesia in recent years. The unidirectional version has a higher increase in net oil palm planted area in Malaysia and Indonesia (71.71% and 117.64%) from 2007 to 2016 than the bi-directional version (46.62% and 105.37%). This is because the unidirectional version is temporally filtered based on the assumption of one-way expansion of oil palm plantation, while the bi-directional version considered the conversion from oil palm to other land cover types (Section 2.3.3).

**3.2 Accuracy assessment**

The mapping performance of AOPD was evaluated first using independent annual oil palm sample set for 2007, 2008, 2009, 2010, 2015 and 2016. The mapping accuracy from the previously developed datasets over Malaysia (Cheng et al., 2019) were also compared. The results of the annual accuracy (F-score) with producer accuracy (PA) and user accuracy (UA) are shown

in Table 3 and 4. PA shows how correctly the reference samples are classified and indicated the omission error (1-PA) while UA represent what percentage of the classes has been correctly classified and is linked with commission error (1-UA). The average annual accuracy for oil palm areas in Malaysia reached 86.22%, which is 8.27% higher than the annual maps from the previous study (Cheng et al. 2019). The improvement of the oil palm mapping performance is mainly due to the different post-processing (one-way expansion and bi-directional oil palm change strategies) and the introduction of the ancillary data (IFL and GMA). Meanwhile, there is no significant difference in the oil palm mapping accuracy among the six years in Malaysia (all above 85% with less than 2% differences, Table 3), indicating the stability and robustness of AOPD. The evaluation using the second annual oil palm sample set in Indonesia shown the average mapping accuracy of 74.20% and the F-score of 0.74 during 2010-2016. The oil palm mapping accuracy was relatively stable during the gap years and the classified years (higher than 72% with 3% fluctuations, Table 4).

Figure 6 shows the direct comparison of the change maps with the images from Google Earth and Landsat, which document the change process. We use time lapse of images when the annual high-resolution images from Google Earth were not available. Here time lapse means the images obtained >1 year intervals. For example, there is no high-resolution images from Google Earth in 2011, so we used the 2010 images as a substitute in Figure 6d and the actual change time is limited within the period (2010-2013). The first three selected regions in Sarawak, Malaysia (Figure 6a and 6b) and Kalimantan Barat, Indonesia (Figure 6c) representing the typical process of oil palm change, i.e. the clearance of primary forest and the replantation of oil palm cultivations. Overall, most of the changes were captured within the range defined by time lapse of the Google Earth images (see the detected change years in the highlighted regions, red shapes). Different from the first three cases (Figure 6a-c), Figure 6d presents another type of oil change from cropland to oil palm in Sumatera Utara, Indonesia.

Our detected change time is also consistent with the timing of change interpreted from Google Earth and Landsat images. The deviation of the detected change years -during 2001-2006 (the grey color) and 2011-2014 (the blue color) from the validation samples (change sample set) is shown in Figure 7. Limited change samples from 2001 to 2006 was collected because of few high-resolution images available during early years. Overall, an agreement between the detected and the actual change time was found in 75.74% of the samples (2/3 of the detected change time matched the actual change time while 1/3 were within a 1-year interval). Further, the change time tended to be more accurate during 2011-2014 (78.20%) compared to 2001-2006 (67.07%) given the constraints by "from-to" type and the range of exact change area of oil palm from 2011 to 2014.

### 3.3 Comparison of our results with statistics and other products

We first compared the oil palm plantation area from our AOPD product with oil palm harvested area from FAO and USDA, and the oil palm plantation area from MPOB (data available from 2011 to 2015) and BPS-Statistics Indonesia (available from 2011 to 2016) (Figure 5). Note that the FAO inventory data for Malaysia from 2011 to 2015 and the USDA statistics from 2011 to 2014 were derived from MPOB (mainly mature area). The FAO statistics included both mature and immature oil palm area during 2011-2013 but only mature oil palm area during 2014-2015, resulting in an abrupt decline in area in the FAO

inventory in 2014 (the orange line in Figure 5(a)). Therefore, the areas from FAO inventory should be used with caution due to the lack of reliable on-field data sources (Ordway et al., 2019).

Compared to FAO and USDA statistics, the annual mean differences from 2001 to 2016 of our results in Malaysia and Indonesia are positive and amount to 2.00 M ha and 1.18 M ha, respectively. The differences were limited to an average of
415 0.08 M ha (FAO) and 0.55 M ha (USDA) in Malaysia but were relatively higher in Indonesia (1.88 M ha compared to FAO and 0.60 M ha compared to USDA), probably because of more confusion from other plantations (i.e., coconuts, rubber and Acacia) and / or more smallholder growth in Indonesia (Lee et al., 2014). There are also small differences of oil palm plantation area in comparison with local national statistics: MPOB (average annual difference of 0.20 M ha) and BPS-Statistics Indonesia (-0.17 M ha). These differences only consist 3.14% and 1.37% of the total oil palm plantation area in 2016 in the two countries.
Trends of oil palm expansion in our mapping results (upper and lower boundary lines) are also compared with statistical data (FAO and USDA from 2001 to 2016, MPOB and BPS-Statistics from 2011 to 2015) (Table S1). Generally, the overall trends of our mapping results (0.758-0.941 M ha/yr) are higher than the FAO (0.561 M ha/yr) and USDA (0.630 M ha/yr) records during the past 16 years, with larger discrepancy in Malaysia (47.07-59.40% higher than FAO and 39.45-53.55% higher than USGS) than Indonesia (16.84-31.68% higher than FAO and 5.99-22.76% higher than USGS). The higher estimation may be
induced by the confusion in other woody plantations such as coconuts and pulp. Although there is high separability between rubber, wattles and palms in PALSAR data (Miettinen and Liew, 2011), the coconuts which belongs to palm trees and have a fan-like shape showed less differences with oil palm compared to other plantations. Another possible reason is the difference in the oil palm plantation definitions (mature and immature oil palm or only mature oil palm included in FAO inventory). Compared to FAO and USDA statistics, increasing trends in our mapping results (0.148-0.178 M ha/yr) are more consistent
with national statistics from MPOB (0.160 M ha/yr) in Malaysia, which include both the mature and immature oil palm during 2011-2015. We should also note that the uni-directional version would have a higher estimation of oil palm plantation area since the assumption of one-way growth. The annual increasing rates of oil palm plantation between our mapping results and other datasets also showed smaller differences in recent period (2011-2015 with national statistics) compared to the whole study period (2001-2016). For example, the increasing oil palm expansion rate of 0.534-0.610 M ha/yr during 2011-2015 in
our product is close to the statistical inventory data, particularly the USDA records (0.536 M ha/yr), while the increasing rate of 0.573-0.674 M ha/yr is relatively higher than USDA (0.520 M ha/yr) and FAO (0.460 M ha/yr) inventory during 2001-2016 in Indonesia. This is also in consistent with the higher uncertainty in the early period and higher reliability in recent years. During the study period, the oil palm export price (total export value/export amount, data source: FAOSTAT) rapidly increased from 402.67 dollars/t in 2006 to the peak (1080.72 dollars/t) in 2011 (Figure S9) but subsequently fell. The crop price is closely
related to demand and may further impact the oil palm market and production (Turner et al., 2011). However, although there is a ~10-20% slowdown of the conversion rate, oil palm plantation area continuously increased after 2011. The land conversion to oil palm may also be affected by multiple factors such as agricultural rent, wages and market-mediated effects (such as tax) (Furumo and Aide, 2017; Taheripour et al., 2019), and the relationship between oil palm expansion and price fluctuation still requires further exploration.

An industrial oil palm plantation dataset developed by a previous study (Gaveau et al., 2016) (Figure 8) was also used to compare our mapping results. The oil palm plantation in Gaveau's dataset was visually interpreted using Landsat datasets in 1973, 1990, 1995, 2000, 2005, 2010 and 2015 in Borneo. The overall distribution of oil palm extent in Borneo are similar between our mapping results (the unidirectional version) and the Gaveau's results (Figure 8a and 8b). The differences were scattered across the whole island with more oil palm plantation areas in our results than in Gaveau's results in the south of

Borneo (Figure 8c, aggregated to proportional maps at 5 km × 5 km to zoom in the difference). Generally, 7.45, 9.23 and 9.86 M ha oil palm plantation area were mapped in AOPD for Borneo during 2010, 2015 and 2016, which is 23.98%, 12.61% and 18.83% larger than the estimates from Gaveau's dataset. Our higher estimation of oil palm plantation area is possibly because some of the smallholder oil palm plantation (1-50 ha in size) is captured in our results whereas only industrial plantations were visually interpreted in Gaveau's results. Misclassification (commission errors) in our results may however also contribute to

our estimation being higher.

The oil palm concession area for Indonesia and Malaysia (Sarawak) for 2014 from global forest watch (www.globalforestwatch.org) is also used in the comparison. This dataset indicated the boundaries of areas allocated by government to companies for oil palm plantation. The oil palm concession area in Indonesia and Malaysia (Sarawak) for 2014 is 12.98 M ha, which is slightly higher (8.7%) than our mapping results (11.85 M ha). However, since the concession data was

compiled from various countries and sources (such as governments and other organizations) with different quality, some location of the existing concessions may be inaccurate (Figure 9(a)) or omitted (Figure 9(b)) compared to our mapping results with PALSAR-2 data. Many concessions are not fully developed (i.e. not planted with oil palm yet) and the number reached more than half of the total 11 M ha (~5.5 M ha) in Sumatra and Kalimantan islands in 2010 (Slette and Wiyono, 2011). Another possible reason for the differences is the inclusion of very small oil palm plantations in our dataset of less than 50 ha, while

most of the oil palm concessions (81.71%) were larger than 1000 ha.

Oil palm expansion is one of the major drivers of deforestation in the studied region (Austin et al. 2018). Therefore, the forest area loss map from Hansen et al. (2013) was overlaid with the AOPD map, and the results are shown for selected areas in Figure 10. in areas (a) and (c), where the year of oil palm expansion is roughly coincides with the year of forest clearance. In other case such as area (b), a larger discrepancy was found in the two maps because of different causes. For example, forest

loss is not always caused by oil palm expansion but timber plantation, logging, fires, conversion from forest to grassland and agriculture (Austin et al., 2018; Kamlun et al., 2016). Meanwhile, expansion of oil palm plantation didn't always occur in forest area, but also in non-forest area. In some regions, the oil palm was planted after the logging of forest immediately (area filled with same color in Figure 10) but in other regions, lands may experience first a forest clearance and then oil palm plantation several years later (indicated by the patches filled with darker color in AOPD than in the forest loss map (Figure

10)). However, the difference of the spatial resolution (30m vs 100m) may also cause some differences, particularly in smallholder and newly developed oil palms. According to our result, 28.20% of total oil palm expansion area overlapped with Hansen's forest loss area (5.38% with the exact same change time, 15.37% later than forest loss year and the remaining 7.46% earlier than the forest loss time). Among the overlapped area, 19.16% of the area has the same change time, 23.67% in 1-year

intervals (may be caused by the time lag between clearance and cultivation), and 38.11% of oil palm expanding areas in AOPD coincide with forest area loss with a lag of at least 2 years. These latter areas may experience first forest clear-cut for other applications or logged and remained unused for several years and then converted to oil palm plantation.

## 4 Discussion

### 4.1 Uncertainty of AOPD

Mapping annual oil palm plantation using remote sensing data in Malaysia and Indonesia is challenging. We developed the first annual oil palm land cover maps (AOPD) from 2001 to 2016 at 100-m resolution combining optical and microwave satellite observations. However, the uncertainties of AOPD, coming from both mapping and change detection, should be acknowledged for the future applications of our dataset. In the mapping procedure, our results showed a good separation between primary forest and oil palm trees but confusion may occur in some impervious area and plantations of other species such as coconuts. As a result, the accuracy of the change detection in the second step was also influenced by the oil palm maps generated from PALSAR/PALSAR-2 data in the first stage. Although oil palm maps for the six years of PALSAR/PALSAR-2 data reached high accuracy at nearly 90% in Malaysia and ~75% in Indonesia, inaccurate inputs in some pixels may lead to cumulative errors in the change detection during the PALSAR data gap years, particularly in Indonesia. The oil palm maps during 2001-2006 without "from-to" inputs, therefore, have more biases compared with the results from 2011 to 2014. Uncertainties could also be induced in the change detection process. Even though the change pixels during the data gap period are constrained by the 100-m oil palm maps from PALSAR before and after that period, the use of moderate resolution MODIS data at 250 m may cause the loss of spatial information and false identification of the change times. Some studies suggested that the fusion of coarse and fine resolution satellite data requires fine resolution images at a certain frequency (Zhang et al., 2017). However, when aiming to conduct consecutive mapping and changes detection, there will always be a trade-off between spatial and temporal resolution (Yin et al., 2018) considering the availability of satellite data such as MODIS and Landsat data (i.e., MODIS has denser observations but coarser spatial resolution than Landsat data). In addition to the satellite data, the change detection algorithm may also bring uncertainties. Because the accuracy of the detected change time by BFAST within a time series is influenced by the signal-to-noise ratio (Verbesselt et al., 2010b), cloud contamination and poor data quality in some regions from MODIS reduced the amount of valid information. And the bias may also be found in the gap years when no breakpoint could be found using BFAST algorithm and the errors were accumulated to years when switching to MODIS before and after PALSAR. However, it is difficult to identify whether the errors are originated from the classification during PALSAR period or the change detection in the gap period. Further improvement could be the use of algorithms which combines the different models (i.e., BEAST) rather than the single-best model (Zhao et al., 2019a). When applying the change detection algorithms, we assumed one-time change in two periods (2001-2007 and 2011-2014). However, multiple changes may occur in the deforestation area when the logging activity is applied first and followed by the replantation of oil palm several years later. More importantly, oil palm will be cut down and replanted after 20 to 25 years for the next rotation in order to make the

maximum profits. This would cause confusion with the transitions between oil palm and other land use types. Therefore, we provided two versions of AOPD: one is the original results with bi-directional oil palm area change, and the other is the unidirectional datasets by assuming all the oil palm loss is from rotation and that a loss is followed by a new oil palm plantation. Despite of these uncertainties, the AOPD annual oil palm maps integrated the strengths of microwave (SAR) and optical satellite observations. SAR has the capability in identifying the oil palm from forest regardless of the weather condition, and MODIS time series has a hyper-temporal density and long-time span. Also, our study gives a good example of integrating fine and coarse datasets. Instead of directly using the coarse dataset, the oil palm maps combined the overall change information for the whole data gap period from fine PALSAR/PALSAR-2 data and the detection of exact change year using coarse MODIS data. In recent years, there is a transition from annual classification to change information mining in remote sensing interpretation to reduce the false changes (Xu et al., 2018b). This method can be used not only in monitoring global oil palm dynamics but also in producing annual land cover maps where only discrete fine resolution observations are available. Since the data scarcity of successive Landsat imagery is common across the world, the algorithm described in this study provides an effective way of combining coarse data to update the annual land cover change. Further, inventory compilation and manual visualization of oil palm change in large extent would remain labour and time consuming (Gaveau et al., 2016; Miettinen et al., 2016; Vijay et al., 2018). Our semi-automatic algorithm in oil palm mapping may thus help to establish a long-term monitoring for oil palm, that can be improved over time with regular validation using ground-based observation or very high-resolution images such as Google Earth.

## 4.2 Applications of AOPD

The 100-m annual oil palm maps from AOPD produced in this study can be used in a number of applications. First of all, it can be readily to be used as a cross-validation reference data for other regional oil palm datasets (e.g., FAO inventory). Second, the annual data can be further used to quantify the spatiotemporal characteristics of oil palm change, estimate the annual oil palm yields, identify the potential oil palm planted area and predict the boundary of oil palm expansion in the future and so on. Overlapping the AOPD with forest maps, peatland maps and other land cover maps can give a clue on how the oil palm expansion influences different ecosystems and their carbon balance. For example, oil palm expansion is the largest single driver of deforestation in Indonesia, which contributed to 2.08 M ha of deforestation (23%) in Indonesia from 2001 to 2016 (Austin et al. 2018). The protected areas were also at long-term risk of deforestation from oil palm cultivation (Vijay et al., 2018). Previous studies revealed that oil palm directly replaced 3.1 M ha (27%) peatland in Peninsular Malaysia, Sumatra and Borneo from 2007 to 2015 (Miettinen et al., 2016), causing the carbon-rich tropical peatland to a strong carbon source (Miettinen et al. 2017a). AOPD at fine spatiotemporal resolution can also serve as land-use change forcing data in the bookkeeping models (Hansis et al., 2015; Houghton and Nassikas, 2017) and possibly dynamic global vegetation models (DGVM) (Sitch et al., 2015) (provided that those models include a specific PFT to represent oil palm (Fan et al., 2015)) to better simulate the carbon emissions and hydrology dynamics. It would improve the carbon budget greatly in Southeast Asia

if DGVMs could systematically simulate biomass, litter and soil carbon changes caused by shifts in oil palm plantation, primary forest, peatlands and fire using accurate and compatible land-use change data.

Another vision lies in the sustainable future of oil palm industry. As the major contributor to the economy that supports thousands of people in the tropical countries, developing oil palm industry has been one of the priorities in these countries (Mahmud et al., 2010; Sayer et al., 2012). At the same time, the possible environmental and ecological consequences of monocultures need to be taken into account for the sustainable development of oil palm industry. For example, Roundtable on Sustainable Palm Oil (RSPO) is established to formulate the standards for the industrial oil palm plantation in South-East Asia,

followed by the foundation of Africa Palm Oil Initiative. Voluntary zero-deforestation commitments in the palm oil industry were also implemented since 2010 (Focus, 2016). However, how many and to what extent large corporations will pay real attention to the rights of local populations remains unknown (Barr and Sayer, 2012).

It is crucial to balance between the rural economic development and environmental protection, especially in the regions with high-biodiversity primary forest and carbon-rich peatlands like Southeast Asia. More complete information on oil palm

plantation (e.g. spatiotemporal changes of oil palm and its consequences) would help to reduce the disputes and provide strategies for oil palm's sustainable development. Our annual oil palm maps would thus contribute to the policy formulation as well as policy evaluation (e.g. national moratorium on new permits for the oil palm conversion from primary natural forests and peat lands (Busch et al., 2015)).

## 5 Data availability

The AOPD in Malaysia and Indonesia from 2001 to 2016 at 100-m resolution are available to the public at https://doi.org/10.5281/zenodo.3467071 (Xu et al., 2019). The dataset includes a set of GeoTIFF images in the WGS_1984_World_mercator projected coordinate system. It can be opened/reprocessed in GIS applications (e.g., QGIS, ArcGIS) and other opening computing environment (R, matlab, etc.). Value 1 represents oil palm while value 0 is Null value.

In this study, we used PALSAR/PALSAR-2 and MODIS NDVI datasets to produce AOPD and SRTM DEM, Intact Forest

Landscape (IFL) and Global Mangrove Atlas (GMA) were used to filter the results in the post-processing. The 25 m resolution PALSAR and PALSAR-2 data provided by Japan Aerospace Exploration Agency (JAXA) from 2007 to 2010 and 2015 to 2016 are available at http://www.eorc.jaxa.jp/ALOS/en/palsar_fnf/data/index.htm after entering basic information. MODIS vegetation index data (MOD13Q1 NDVI) collection 6 (250m) from 2000 to 2015 and SRTM DEM (30 m) were obtained from the Land Processes Distributed Active Archive Center (https://lpdaac.usgs.gov/). IFL is available from

http://www.intactforests.org/ and GMA can be downloaded from http://geodata.grid.unep.ch/results.php).

## 6 Conclusion

Combining the optical and microwave satellite observations, we developed the first annual oil palm maps (AOPD) in Malaysia and Indonesia from 2001 to 2016 at 100-m resolution using the image classification and change detection analysis. The dataset reached a high accuracy in both annual classification and change-detection. As a result, this dataset provided insights and
details on dynamic oil palm changes for Malaysia and Indonesia from the perspective of remote sensing and can serve as a supplement for statistics. Further applications of the dataset include but is not limited to regional carbon studies, water and agricultural management, biodiversity and conservation protection and the sustainable development of oil palm industry. The annual updating method in this study that fully used information from discrete fine resolution data and continuous coarse resolution data is also expected to be applicable in other regions facing data scarcity.

**Acknowledgements**

This research was partially supported by the National Key R&D Program of China (grant number: 2017YFA0604401).

**Competing interests**

The authors declare that they have no conflict of interest.

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

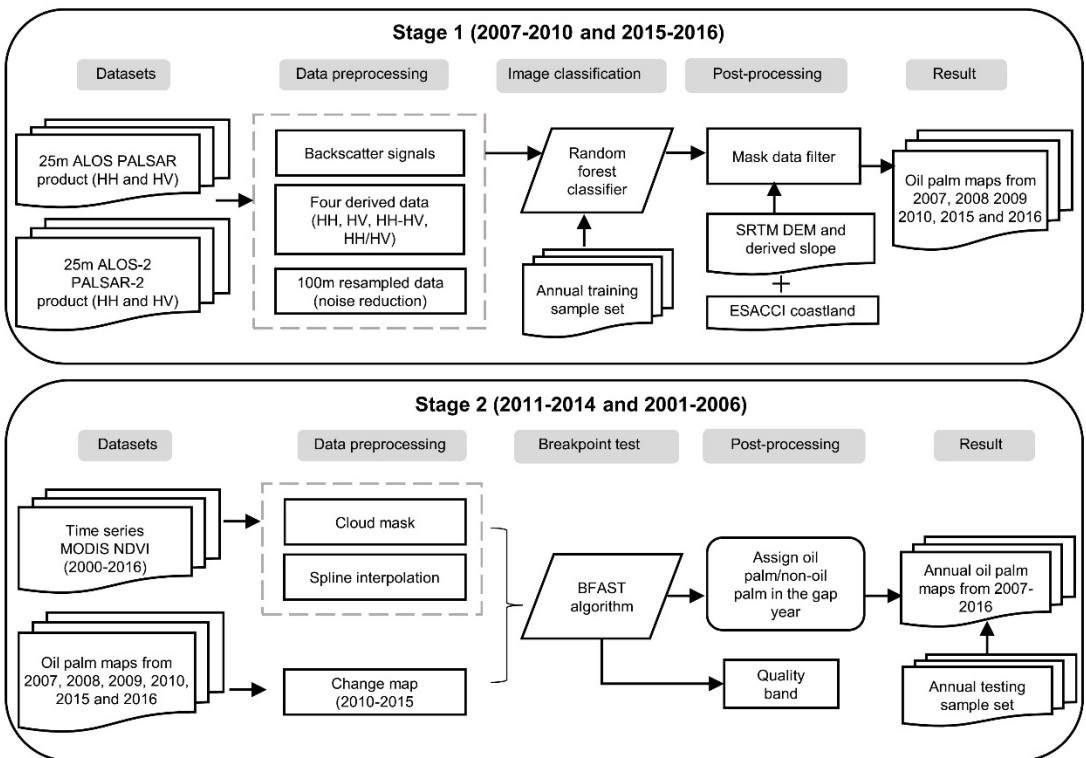

**Figure 1: Workflow of the annual oil palm mapping procedure. Stage 1 stands for oil palm mapping using PALSAR/PALSAR-2 data, and Stage 2 stands for change-detection based oil palm updating using MODIS NDVI.**

**Table 1: The distribution of training data (unit: pixel). Malay.: Malaysia. Indon.: Indonesia.**

|  | Oil palm | | Other vegetation | | Water | | Others | | Total | |
|---|---|---|---|---|---|---|---|---|---|---|
|  | Malay. | Indon. | Malay. | Indon. | Malay. | Indon. | Malay. | Indon. | Malay. | Indon. |
| 2007 | 1,228 | 2,368 | 2,970 | 3,351 | 570 | 762 | 185 | 1,323 | 4,953 | 7,804 |
| 2008 | 1,279 | 1,921 | 2,994 | 3,561 | 570 | 818 | 185 | 1,039 | 5,028 | 7,339 |
| 2009 | 1,387 | 2,065 | 3,179 | 3,893 | 570 | 842 | 185 | 1,161 | 5,321 | 7,961 |
| 2010 | 1,405 | 2,005 | 3,228 | 3,824 | 570 | 837 | 185 | 1,076 | 5,388 | 7,742 |
| 2015 | 1,475 | 2,349 | 3,430 | 4,287 | 570 | 656 | 185 | 1,360 | 5,660 | 8,652 |
| 2016 | 1,475 | 2,312 | 3,430 | 4,020 | 570 | 562 | 185 | 1,253 | 5,660 | 8,147 |

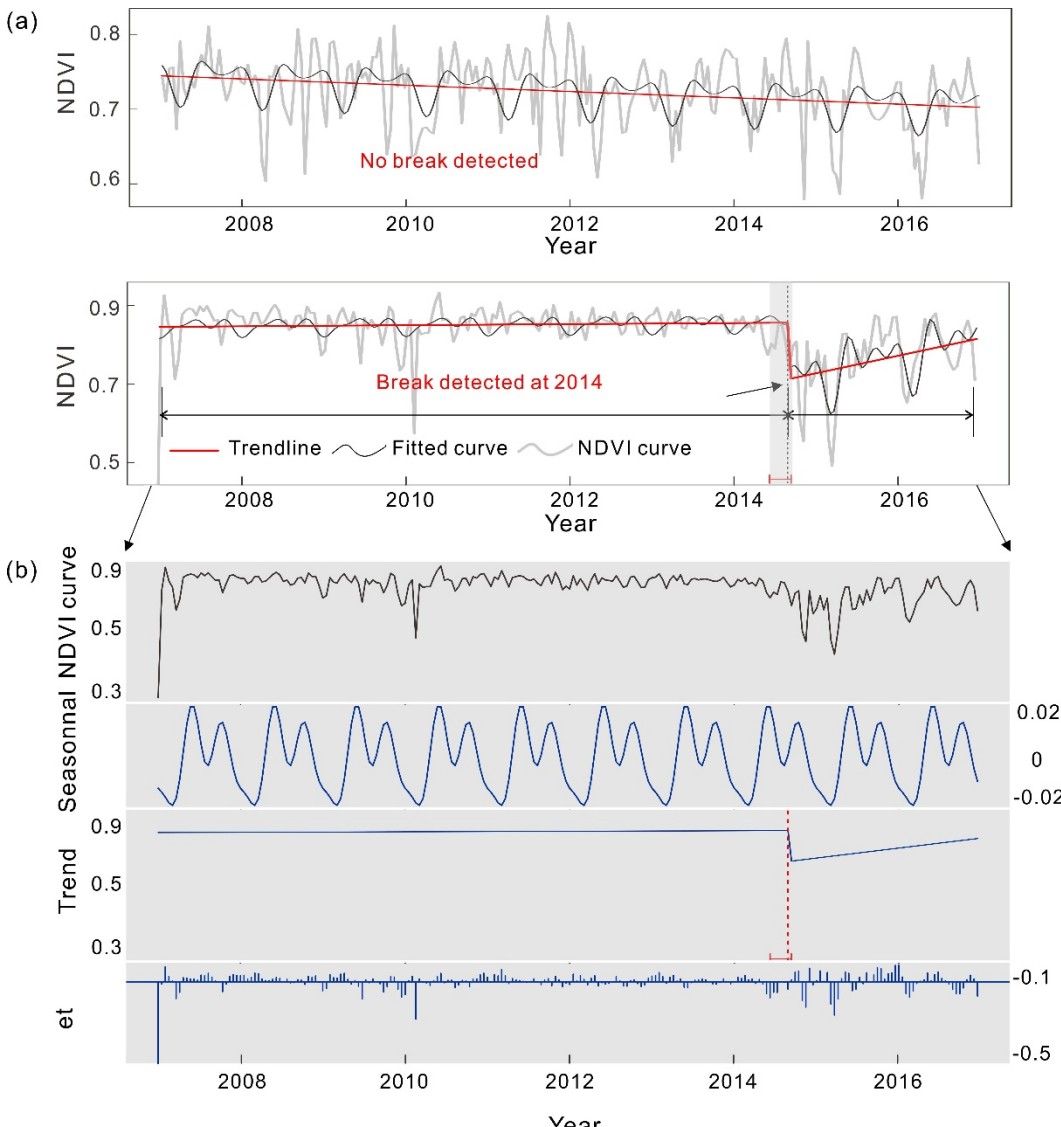

Figure 2: Examples of the breakpoint detection in the MODIS time series using the BFAST algorithm. (a) The two cases present when the algorithm is able to detect the break in the NDVI time-series. The NDVI curve is the original 16-day composite MODIS NDVI time series. The fitted curve is the pre-processed NDVI after cloud masking and spline interpolation. Trendline shows the fitted trend for each segment after seasonal-trend decomposition using BFAST. (b) The seasonal-trend decomposition of the 16-day NDVI time series using BFAST for the second example. The algorithm decomposes the time series into three components: trend, seasonality, and residuals (et).

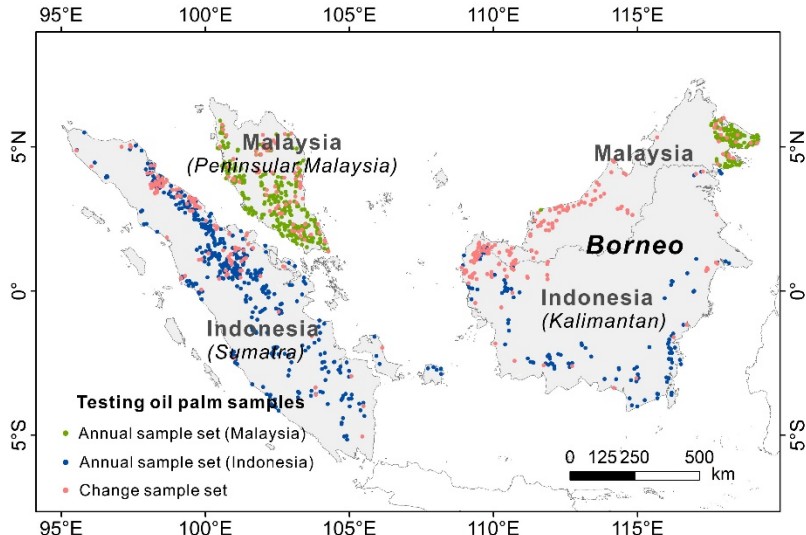

**Figure 3: Spatial distribution of oil palm samples in the two validation datasets. The annual sample set contains 2986 (in 2016) samples in Malaysia which were interpreted for 2007, 2008, 2009, 2010, 2015 and 2016 and 7667 (in 2016) samples in Indonesia interpreted from 2010-2016. These samples were used to validate the annual maps developed from PALSAR/PALSAR-2 data. Of the annual sample set in Malaysia, oil palm samples consist of 16.92% (505) while the forest, water and others consist of 78.16%, 2.48% and 2.44%, respectively. The Indonesian annual sample set contains 601 (7.84%) oil palm samples and the rest (92.16%) were other types. The change sample set includes 370 oil palm samples which were converted in the interpolated period (2001-2006 and 2011-2014). This sample set, with change year labelled, is used to assess the change detection result in the gap years.**

**Table 2: The distribution of annual validation sample set for Malaysia and Indonesia (unit: pixel).**

| | Malaysia | | | | | | Indonesia | | |
|---|---|---|---|---|---|---|---|---|---|
| | Oil palm | Other vegetation | Water | Others | Total | | Oil palm | Not oil palm | Total |
| 2007 | 371 | 2,335 | 68 | 74 | 2,848 | 2010 | 547 | 7,066 | 7,613 |
| 2008 | 398 | 2,334 | 71 | 76 | 2,879 | 2011 | 559 | 7,063 | 7,622 |
| 2009 | 418 | 2,335 | 71 | 76 | 2,900 | 2012 | 568 | 7,068 | 7,636 |
| 2010 | 433 | 2,335 | 71 | 76 | 2,915 | 2013 | 575 | 7,078 | 7,653 |
| 2015 | 505 | 2,336 | 75 | 76 | 2,992 | 2014 | 588 | 7,072 | 7,660 |
| 2016 | 505 | 2,334 | 71 | 73 | 2,983 | 2015 | 594 | 7,073 | 7,667 |
| | | | | | | 2016 | 601 | 7,066 | 7,667 |

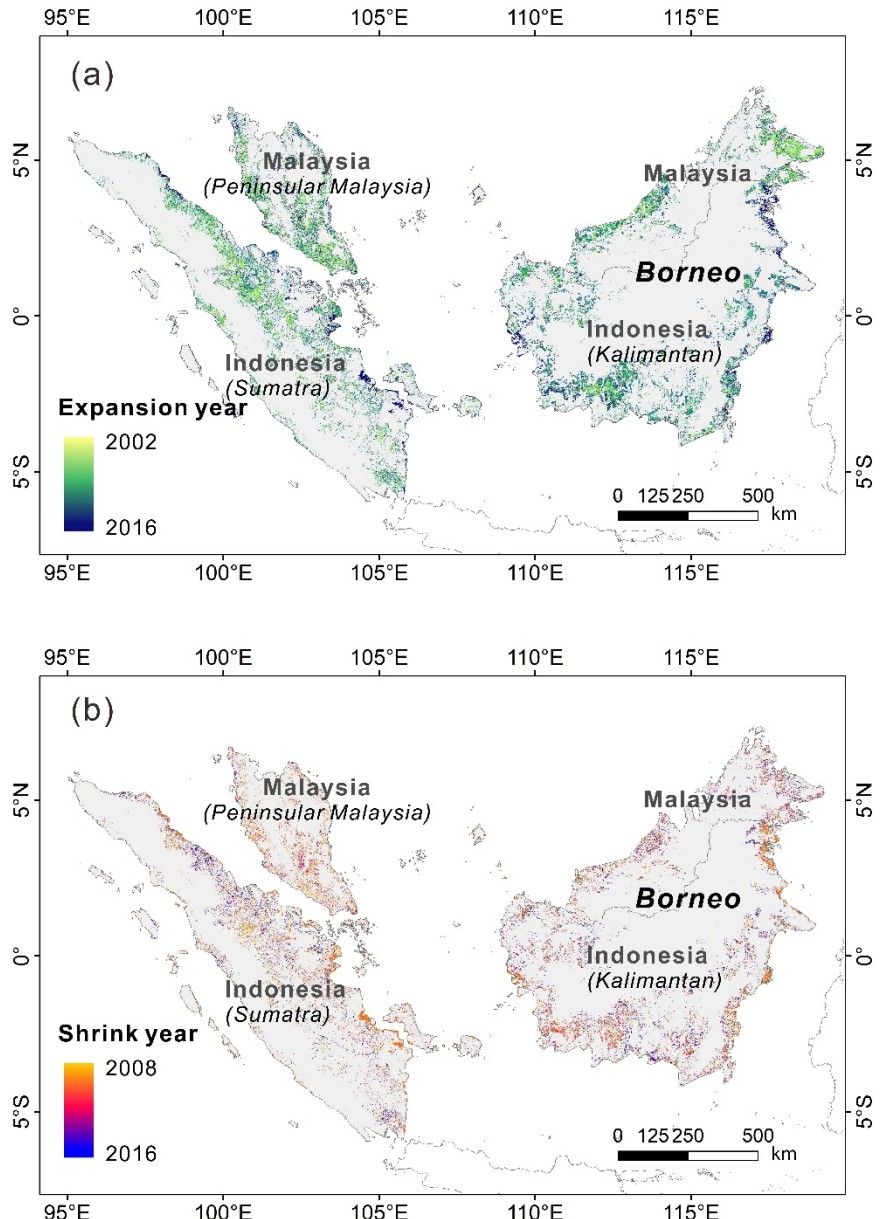

Figure 4: Year of oil palm change at 100m resolution in the study area from 2002 to 2016. a) expansion, 2002-2016, b) shrinkage,
2008-2016. During 2011-2014, the "from-to" types of the change pixels were pre-defined in the 2010 and 2015 land cover maps
derived from PALSAR and PALSAR-2 data, respectively. Therefore, both the expansion and shrinkage year of oil palm were
available in this period using the change-detection method. During 2001-2006, the oil palm distribution of the start year is unknown.
Here we assumed one-way expansion of oil palm before 2007 and adopted the change-detection algorithms in the 2007 oil palm
extent. Thus, the expansion year was traced back to 2002. The grey background refers to the study area.

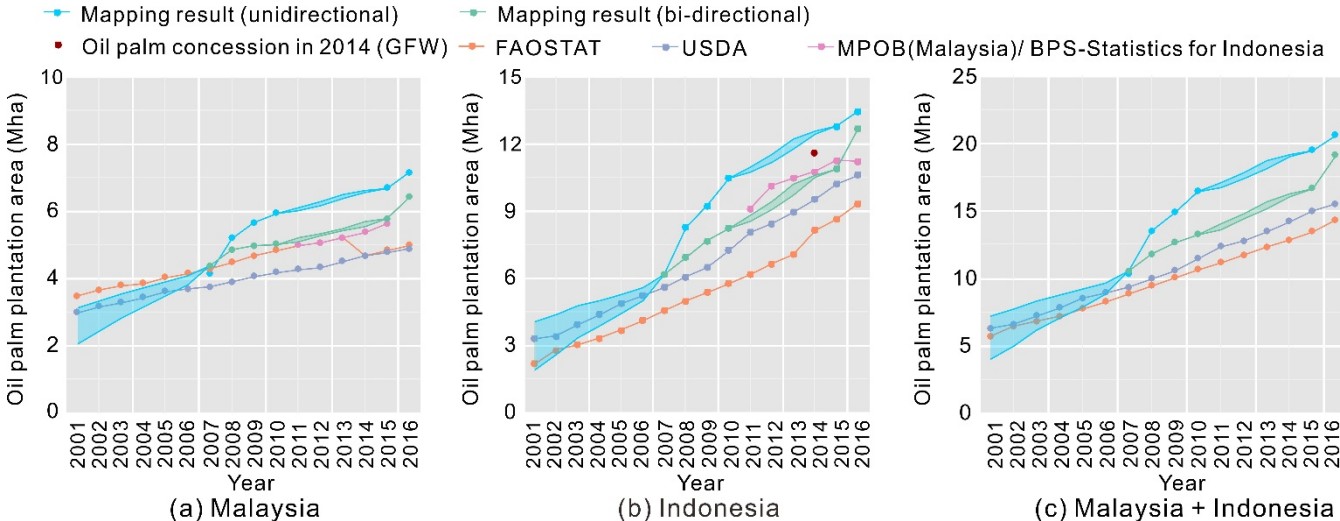

**Figure 5: Comparison of the annual oil palm plantation area among FAO and USDA statistics, MPOB records for Malaysia, BPS-Statistics and oil palm concessions from GFW for Indonesia and our mapping results in a) Malaysia, b) Indonesia and c) Malaysia and Indonesia from 2001 to 2016. The blue lines represent the gross gain (unidirectional expansion) while the green lines show the net changes of oil palm from 2007 to 2016. The shaded area within the two boundary lines are the uncertainty range of the oil palm area. The upper boundary lines represent the upper limit area of oil palm within the two periods (2011-2014 and 2001-2006), whereas the lower boundary lines are the lower limit according to our results. Note that during the gap between the two periods, no uncertainty could be derived, which does not mean that the uncertainty was small.**

**Table 3. The comparison of the oil palm accuracy between our mapping results and Cheng et al. (2019) for the six mapping years in Malaysia. UA: User's Accuracy; PA: Producer's Accuracy**

| Year | Cheng et al (2019) | | | Our results | | |
|------|---------|--------|--------|---------|--------|--------|
| | *F*-score | UA (%) | PA (%) | *F*-score | UA (%) | PA (%) |
| 2007 | 0.74 | 78.02 | 70.63 | 0.86 | 93.40 | 80.05 |
| 2008 | 0.78 | 82.5 | 73.83 | 0.88 | 93.22 | 82.91 |
| 2009 | 0.75 | 79.76 | 71.13 | 0.86 | 92.12 | 81.10 |
| 2010 | 0.79 | 80.92 | 77.02 | 0.85 | 93.89 | 78.06 |
| 2015 | 0.83 | 80.31 | 85.25 | 0.86 | 92.08 | 80.59 |
| 2016 | 0.79 | 78.5 | 79.13 | 0.86 | 87.47 | 84.36 |

**Table 4. The oil palm accuracy in Indonesia from 2010-2016. UA: User's Accuracy; PA: Producer's Accuracy**

| Year | Our results | | |
|------|---------|--------|--------|
| | *F*-score | UA (%) | PA (%) |
| 2010 | 0.75 | 69.47 | 74.95 |
| 2011 | 0.75 | 70.38 | 74.83 |
| 2012 | 0.75 | 71.48 | 75.05 |
| 2013 | 0.75 | 72.39 | 74.79 |
| 2014 | 0.74 | 72.58 | 74.28 |
| 2015 | 0.72 | 68.46 | 71.83 |
| 2016 | 0.72 | 69.97 | 72.33 |

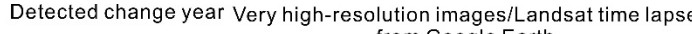

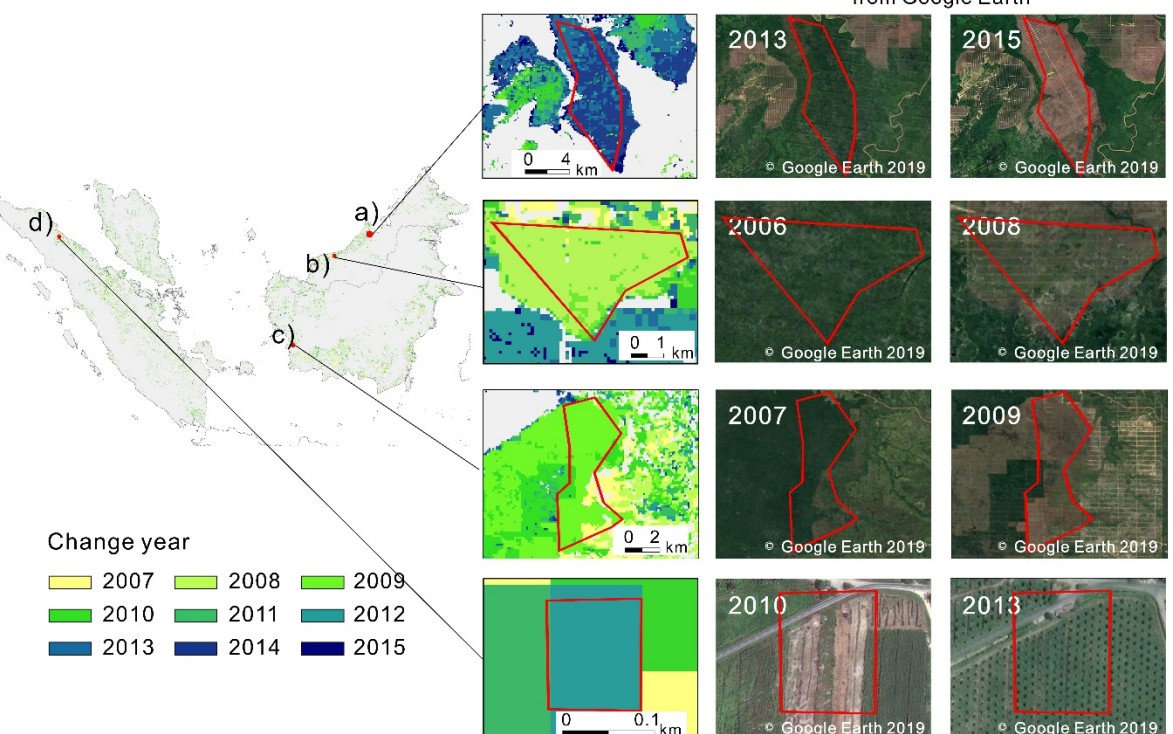

**Figure 6: Visual comparison of the detected change years with the high-resolution images and medium-resolution Landsat images from Google Earth. The color of the first column represents the change detected time in our results. The red shape highlights the change areas. a) and b) are two selected regions located in Sarawak, Malaysia, the Landsat images in the right indicate that the deforestation and plantation of oil palm occurred between 2013 and 2015, 2006 and 2008, respectively, and the change times (2014 and 2008) were captured in the result maps; c) is an example of change detected in 2009 in Kalimantan Barat, Indonesia, where forest type is presented in the Landsat images in 2007 and oil palm plantation shown in 2009; d) is a case showing the conversion of cropland to oil palm in Sumatera Utara, Indonesia according to the high-resolution images from Google Earth. The young oil palm trees in the 2013 image indicate that the conversion may have occurred in one or two years before, which matched the results in our maps (detected change time in 2012).**

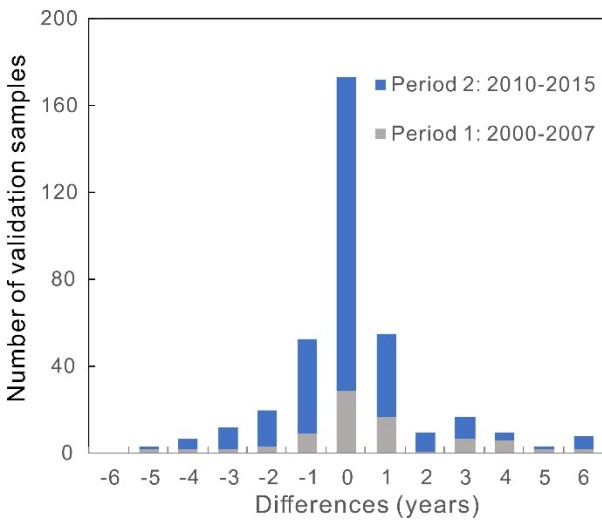

**Figure 7: Difference between the detected change years using MODIS NDVI dataset and the exact change years from the reference dataset (Google Earth and Landsat). Negative values in x-axis refer to the detected year earlier than the actual change year.**

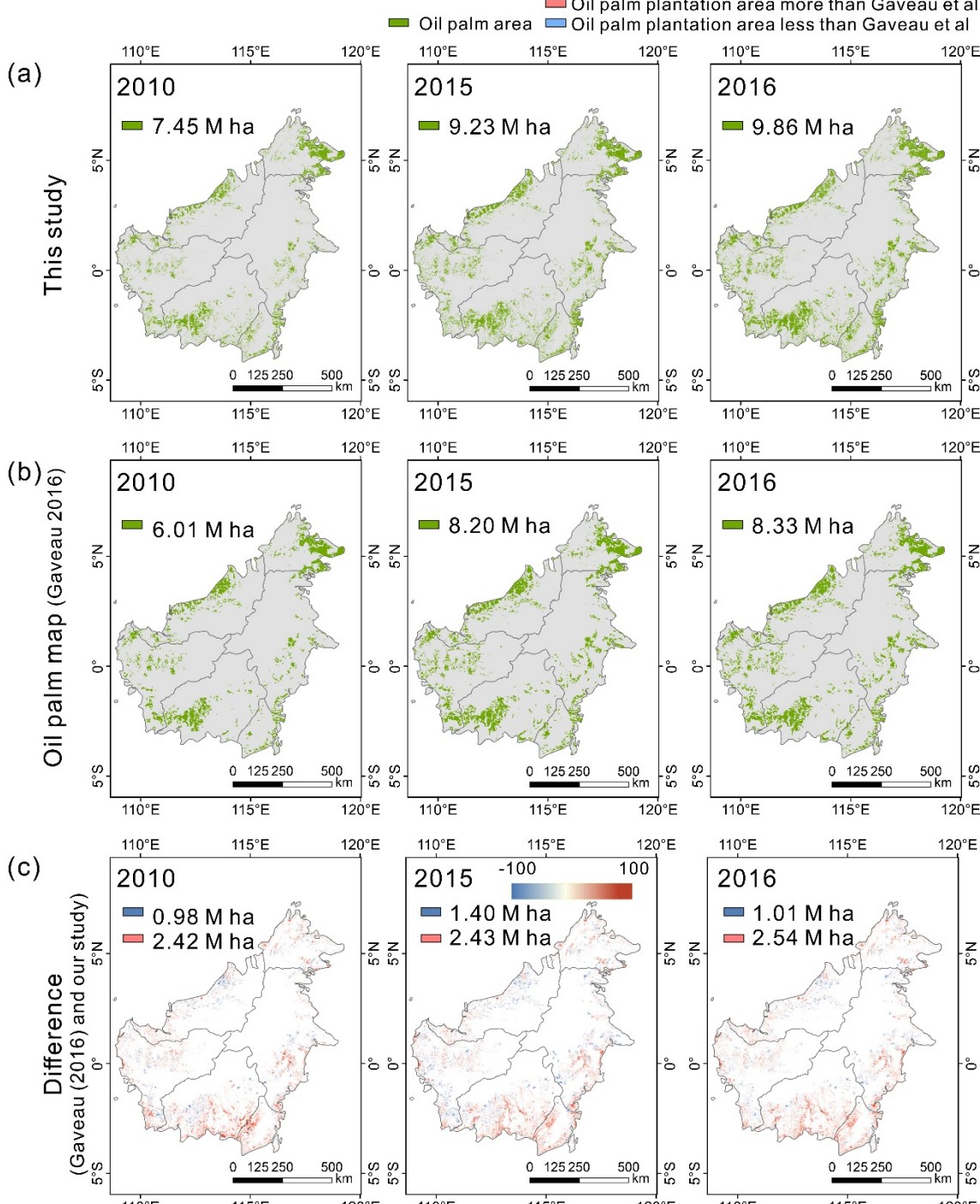

Figure 8: Comparison with existing oil palm datasets in Borneo (Gaveau et al. 2016) for year 2010, 2015 and 2016. The oil palm maps were aggregated to proportional maps at 5 km × 5 km to visualize the difference in the third rows.

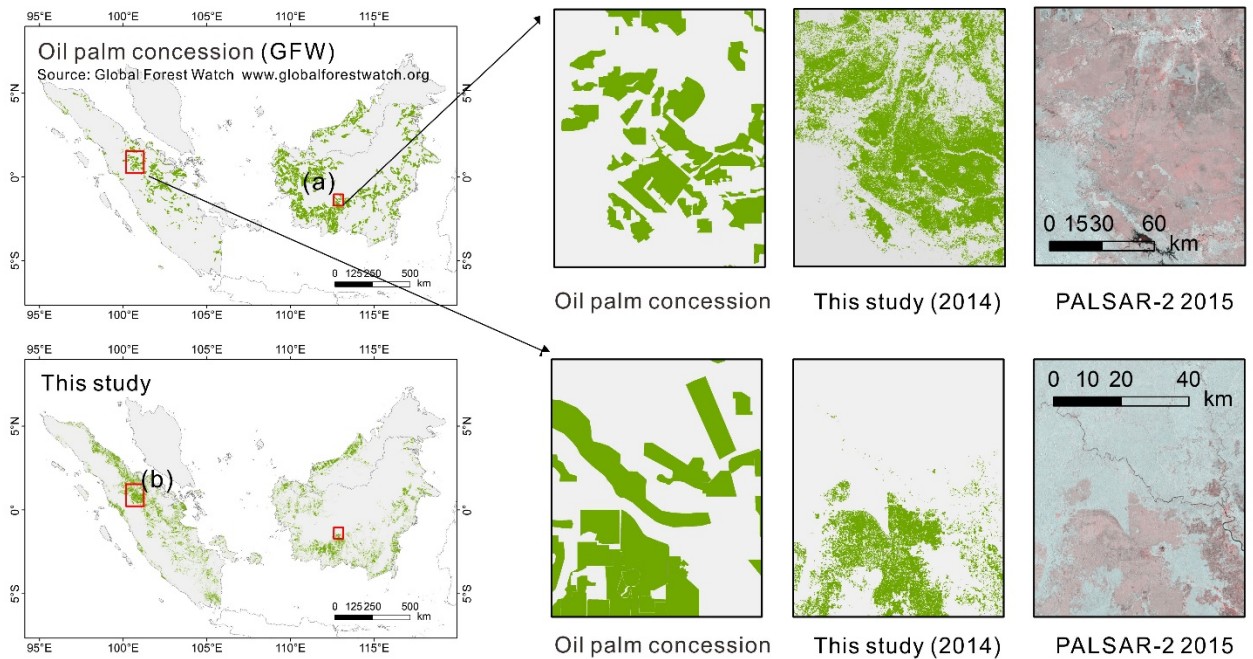

**Figure 9: Comparison with oil palm concession from Global forest watch (GFW) for year 2014. The PALSAR-2 images were**
870 **composited in RGB format (HH, HV, HV).**

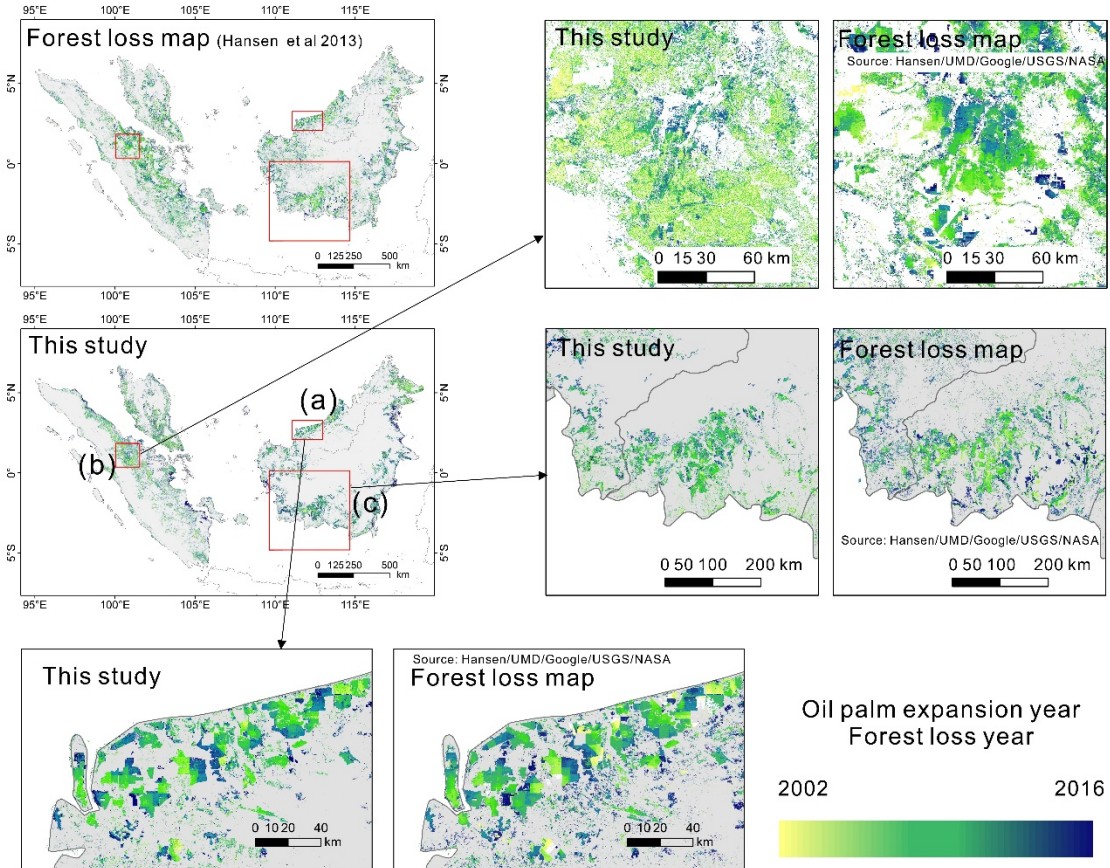

**Figure 10: Comparison of oil palm expansion map in this study with the Landsat forest area loss map (Hansen et al. 2013).**

875