# Peer review of "Annual oil palm plantation maps in Malaysia and Indonesia from 2001 to 2016"

_Earth System Science Data, 2019_

## Referee Comment (RC1) · Anonymous Referee #1 · 21 Nov 2019

The article presented the first annual oil palm plantation maps in Malaysia and Indonesia and demonstrate the accuracy of the maps through various comparisons with existing statistic dataset and regional maps. It's an interesting paper that exhibits the efficiency of fusing optical and radar data in over coming data gaps to produce consistent annual maps. However, there are quite a few details in the abstract and introduction session that need to be checked. Some statement are lacking adequate references. More detail needs to be given on the methods, especially validation approach. Some of the conclusions in the discussion section need to be backed up, either by reference or by results. I'm not very convinced by the results due to limited information was given to the independence validation approach.

Please see the particular comments below: Abstract/Introduction:  c 12: The land

[Figure]

convention to oil palm plantations not always lead to deforestation. âǍć 26: Current discussion is not strong enough to support the conclusion that the higher trend in this study is due to the inclusive of smallholder farmers. (more comments in the Results part, section 3.3) âǍć 36: Corley, 2009- any more recent ref to support the expected growing rate from 2003? âǍć 38:"forest cover dropped from 76% to 9% since 1990 in Malaysia and Indonesia". Please double check these numbers, and cross reference with other sources. âǍć 43: There are quite a few existing dataset/report that are providing continuous information about the expansion of oil palm in Indonesia and Malaysia. E.g. https://theicct.org/sites/default/files/publications/ICCT_palm-expansion_Feb2012.pdf âǍć 59: There are quite a lot of Machine learning or Deep Learning based methods for automatic identification of oil palms.

Methods: âǍć Any co-registration issue between MODIS and ALOS/ALOS2? âǍć 149: Any other prove that no calibration is needed between ALOS and ALOS2 in Indonesia and Malaysia? The study site for the two referenced papers are not for these two countries specifically (Thus with different incident angel, weather condition, etc). âǍć 108: How dose 98.91% been calculated? âǍć Why the NDIV information from MODIS is not used as input to the RF model for classification? âǍć 213: How many MODIS time series are used exactly? How many are actual data and how many are interpolated? As the author explained, Indonesia and Malaysia are heavily affected by clouds, so as MODIS NDVI as well. âǍć Eq 4, 5 and 6: some errors in explanation. âǍć More information is needed for the validation methods (2.5). E.g how many samples are there for each land use class for each year? âǍć 726: Fig 3: are all the 2986 annual sample from 2016, and other years are interpolated? And how? âǍć 725: Fig 3: the distribution of validation dataset is very uneven. There is no annual sample set in Sumatra Indonesia at all. âǍć 300: How does the total number of validation points (5000) been decided? What's the ratio of the validation points to the total pixel been detected as change?

Results: âǍć Paragraph 1 and 2: There is no other information/ref/map/graph/table

provided to support many of the conclusions in these two paragraphs. Some of the sentences read like discussion rather than results. • Section 3.3: Have you compared your results from Global forest watch, oil palm concession dataset 2014? • Section 3.3: There lacks adequate reference to support the linkage between oil palm expansion, price fluctuation. • Section 3.3: There are potentially more reasons to explain the higher estimated oil palm area in this study compared to existing dataset. More evidence is needed to exclude other reasons and draw the conclusion to smallholders' oil palm plantation. Especially the minimum mapping unit in this paper is 1ha. • 435: what does 'limited bands in ALOS/ALOS 2 mean?

---

## Referee Comment (RC2) · Anonymous Referee #2 · 30 Nov 2019

The manuscript is addressing the annual oil palm mapping in Malaysia and Indonesia from 2001 – 2016 by using PALSAR/PALSAR-2 imagery, and fill the PALSAR data gap (2011-2014) by using the MODIS data and the BFAST method. This study is well designed and the paper is very well written. But some parts should be further improved before its consideration for publication. Here are my comments for consideration:

1. Effects of stand age. How the stand age could affect the identification of the oil palm plantation as well as the robustness of the BFAST approach? This study claims that the maps include young oil palm trees and smallholder oil palm plantations. What strategies have been considered to make sure the inclusion of young trees and smallholder plantations? 2. Effects of multiple data resolutions. Why does the resolution of 100m perform better to estimate oil palm planting area, not the 50m or other resolution? Is

resolution of 100m sufficient to depict the smallholder details? Which resample technique did you use to resample 25-m PALSAR to 100-m? How did you integrate your 100-m oil palm maps with the 250-m land cover change maps? 3. How many types of land cover were got with the RF classification? Is the multi-class classification consistent with Table 1? Or the binary classification (oil palm; non-oil palm)? 4. You provided two version of oil palm datasets: one considers the oil palm expansion (unidirectional change) and the other one considers oil palm shrinkage (bi-directional change). Which version is more consistent with statistics? Which version is more accurate based on your validation samples? In Figure 5, the oil palm change in 2001-2007 is also unidirectional, thus the color of line might be blue, not green. 5. If there were more than one change time in 2011-2014 or 2001-2006, how did you allocate land cover types?

---

## Author Comment (AC1) · 31 Dec 2019

**Response to comments**

Paper #: essd-2019-137 Title: Annual oil palm plantation maps in Malaysia and Indonesia from 2001 to 2016 Journal: Earth System Science Data

**Reviewer #1:**

**General Comments:**

**Comment #1**

The article presented the first annual oil palm plantation maps in Malaysia and Indonesia and demonstrate the accuracy of the maps through various comparisons with existing statistic dataset and regional maps. It's an interesting paper that exhibits the efficiency of fusing optical and radar data in over coming data gaps to produce consistent annual maps. However, there are quite a few details in the abstract and introduction session that need to be checked. Some statement are lacking adequate references. More detail needs to be given on the methods, especially validation approach. Some of the conclusions in the discussion section need to be backed up, either by reference or by results. I'm not very convinced by the results due to limited information was given to the independence validation approach.

**Response #1**

We thank the reviewer for the comments and suggestions. Please see the detailed point-by-point responses below.

**Specific Comments:**

**Comment #1**

Abstract/Introduction: Line 12: The land convention to oil palm plantations not always lead to deforestation.

**Response #1**

Oil palm conversion takes places not only in forest but also agroforests, agricultural fallows, bare lands and etc. So we changed the original sentence to "The land conversion to oil palm plantations poses risks to deforestation (50% of the oil palm was taken from forest during 1990-2005, Koh and Wilcove, 2008), loss of biodiversity, and greenhouse gas emission over the past decades." (Abstract, Lines 12-14).

**Comment #2**

Line 26: Current discussion is not strong enough to support the conclusion that the higher trend in this study is due to the inclusive of smallholder farmers. (more comments in the Results part, section 3.3)

**Response #2**

We totally agree with this. The inclusive of smallholder farmers is one of the potential reasons of the higher trend in this study. We rewrote the conclusions and excluded it in the abstract (Abstract, Lines 26-28): "The higher trends from our dataset are consistent with those from the national inventories with limited annual average difference in Malaysia (0.2 M ha) and Indonesia (-0.17 M ha)." And we also discussed more possible reasons in the Result and discussion part (Please see the reply to comment#19).

**Comment #3**

Line 36: Corley, 2009- any more recent ref to support the expected growing rate from 2003?

**Response #3**

We updated the growing rate of oil palm fruit production in Malaysia and Indonesia to 2017 according to FAO statistics and added a new reference projecting a considerable expansion of oil palm cultivation worldwide in the future in **Section 1, Lines 35-37** :"According to the Food and Agriculture Organization (FAO), Malaysia and Indonesia account for 81.90% of the global oil palm fruit production in 2017, an increase by 179.72% from 2000 to 2017 (see http://faostat.fao.org) that is projected to continue in the future (Murphy, 2014). "

**Reference:**

Murphy, D. J. (2014). The future of oil palm as a major global crop: opportunities and challenges. J Oil Palm Res, 26(1), 1-24.

**Comment #4**

Line 38:"forest cover dropped from 76% to 9% since 1990 in Malaysia and Indonesia". Please double check these numbers, and cross reference with other sources.

**Response #4**

Sorry for the mistake. The peat swamp forest dropped from 76% to 29% since 1990 in Malaysia and Indonesia according to the reference. We also added references to show the deforestation caused by oil palm expansion on Section 1 Lines 38-41:" In Malaysia and Indonesia, more than 50% of the oil palm plantation was converted from forest during 1990-2005 (Koh and Wilcove, 2008) and industrial plantations dominated by oil palm (72.5% of all plantations) caused a 60% decrease of peatland forest from 2007 to 2015 (Miettinen et al., 2016)."

**Comment #5**

Line 43: There are quite a few existing dataset/report that are providing continuous information about the expansion of oil palm in Indonesia and Malaysia. E.g. https://theicct.org/sites/default/files/publications/ICCT\_palm- expansion\_Feb2012.pdf

**Response #5**

We thank the reviewer for this information. We added the references of the continuous mapping of oil palm on **Section 1, Lines 50-52**: "The continuous mapping of oil palm on peatland in 1990, 2000, 2007 and 2010 described the dynamic change of oil palm on peat during the past 30 years (Miettinen et al., 2012)." Here we also modified the text from continuous to annual mapping in **Section 1 Line 45**: "However, annual information on the expansion of oil palm plantations is poorly documented in Malaysia and Indonesia."

**Comment #6**

Line 59: There are quite a lot of Machine learning or Deep Learning based methods for automatic identification of oil palms.

**Response #6**

We added the recent deep learning based automatic identification references here as suggested on **Section 1, Line 63-66**: "3) interpretation methods from manual to semi- and fully automatic identification (Baklanov et al., 2018; Cheng et al., 2019; Li et al., 2017a; Mubin et al., 2019; Ordway et al., 2019), 4) products going from oil palm land cover maps to more detailed datasets on plantation structure, e.g. tree counting (Li et al., 2019; Cheang et al., 2017)."

**Reference:**

Baklanov, A., Khachay, M., and Pasynkov, M.: Application of fully convolutional neural networks to mapping industrial oil palm plantations, International Conference on Analysis of Images, Social Networks and Texts, 2018, 155-167,

- Cheang, E. K., Cheang, T. K., and Tay, Y. H. J. a. p. a.: Using convolutional neural networks to count palm trees in satellite images, 2017.
- Li, W., Fu, H., Yu, L., and Cracknell, A. J. R. S.: Deep learning based oil palm tree detection and counting for high-resolution remote sensing images, Remote Sensing, 9, 22, 2017a.
- Mubin, N. A., Nadarajoo, E., Shafri, H. Z. M., and Hamedianfar, A.: Young and mature oil palm tree detection and counting using convolutional neural network deep learning method, Int. J. Remote Sens., 40, 7500-7515, 10.1080/01431161.2019.1569282, 2019.

**Comment #7**

**Methods:**

Any co-registration issue between MODIS and ALOS/ALOS2?

**Response #7**

We've checked there is no co-registration issues. And other analysis was also directly conducted in MODIS and PALSAR data in previous researches (Qin et al., 2017 and Zhang et al., 2019). We will further clarify this point in the revised manuscript.

**Reference:**

- Zhang, Y., Ling, F., Foody, G. M., Ge, Y., Boyd, D. S., Li, X., Du, Y., and Atkinson, P. M.: Mapping annual forest cover by fusing PALSAR/PALSAR-2 and MODIS NDVI during 2007–2016, Remote Sens. Environ., 224, 74-91, https://doi.org/10.1016/j.rse.2019.01.038, 2019.
- Qin, Y., Xiao, X., Dong, J., Zhou, Y., Wang, J., Doughty, R. B., Chen, Y., Zou, Z., and Moore, B.: Annual dynamics of forest areas in South America during 2007–2010 at 50-m spatial resolution, Remote Sens. Environ., 201, 73-87, https://doi.org/10.1016/j.rse.2017.09.005, 2017.

**Comment #8**

Line 149: Any other prove that no calibration is needed between ALOS and ALOS2 in Indonesia and Malaysia? The study site for the two referenced papers are not for these two coun-tries specifically (Thus with different incident angel, weather condition, etc).

**Response #8**

We randomly generated 250,000 points using ArcGIS 10.3 in our study area and compared the HH/HV values of these points during the 6 years following Qin et al (2016) and Cheng et al (2019)'s practice (**Figure S2**, reproduced below). According to the histogram, the backscattering value of PALSAR/PARSAR-2 are relatively stable in the study period. The reference which presented the stability of annual PALSAR/PALSAR-2 HH and HV values in Malaysia was also added (Cheng et al., 2019). Meanwhile, the HH and HV values for oil palm and forest is also shown in **Figure S3** (reproduced below) and indicate the separability between the two land cover types for both PALSAR/PALSAR-2 data. We will add these points in the revised manuscript. We produced the classification map using the training samples from each corresponding year, the influence of calibration differences between PALSAR/PALSAR-2 data will not influence the mapping results.

**Reference:**

- Cheng, Y., Yu, L., Xu, Y., Lu, H., Cracknell, A. P., Kanniah, K., and Gong, P.: Mapping oil palm plantation expansion in Malaysia over the past decade (2007–2016) using ALOS-1/2 PALSAR-1/2 data, Int. J. Remote Sens., 1-20, 2019
- Qin, Y., Xiao, X., Dong, J., Zhou, Y., Wang, J., Doughty, R. B., Chen, Y., Zou, Z., and Moore, B.: Annual dynamics of forest areas in South America during 2007–2010 at 50-m spatial resolution, Remote Sens. Environ., 201, 73-87, https://doi.org/10.1016/j.rse.2017.09.005, 2017.

**Figure S2** Density distribution of PALSAR/PALSAR-2 (a) HH (dB) and (b) HV (dB) in study area for 2007, 2008, 2009, 2010, 2015 and 2016 based on 250000 randomly generated points. The mean and standard deviation (std) value for the six years were given (mean: -7.44~-6.98 of HH and -13.47~-13.01

of HV; std: 2.52~2.90 of HH and 3.05~3.76 of HV). According to the result, the backscatter signals are relatively stable for the given period (2007–2010 and 2015–2016).

**Figure S3** Comparison between PALSAR/PALSAR-2 (a) HH (dB) and (b) HV (dB) for forest and oil palm based on the training points. The HV (dB) for the forest and oil palm samples are differentiable during the given period (2007–2010 and 2015–2016).

**Comment #9**

Line 108: How dose 98.91% been calculated?

**Response #9**

We updated the number (96%) according to the reference (Petrenko et al., 2016) on Section 2. 1, Line 113-115: "Thus, we chose as a study area the whole Malaysia, Sumatra and Kalimantan in Indonesia, encompassing 96% of the total oil palm production in Indonesia (Petrenko et al., 2016)."

**Reference:**

Petrenko, C., et al. (2016). "Ecological impacts of palm oil expansion in Indonesia." J Washington : International Council on Clean Transportation.

**Comment #10**

Why the NDIV information from MODIS is not used as input to the RF model for classification?

**Response #10**

The use of coarse resolution MODIS information in RF may negate the benefits of our classification based on higher spatial resolution PALSAR data, keeping in mind that the change detection results during the gap years is based on the results from that classification. Second, we also found that the spectral information used to derive NDVI is quite similar between a tropical forest and a mature oil palm plantation, which induces confusion in the classification (Razak., 2018). Some studies used the fusion method (such as super-resolution mapping) to fusing coarser resolution MODIS with higher resolution PALSAR data, but these algorithms require large computational cost and were always applied to small scenes. For these two reasons, we didn't include the MODIS NDVI in the RF model. We will further add these points in the revised manuscript.

**Reference:**

Razak, J. A. B. A., Shariff, A. R. B. M., Ahmad, N. B., & Ibrahim Sameen, M. (2018). Mapping rubber trees based on phenological analysis of Landsat time series data-sets. Geocarto international, 33(6), 627-650.

**Comment #11**

Line 213: How many MODIS time series are used exactly? How many are actual data and how many are interpolated? As the author explained, Indonesia and Malaysia are heavily affected by clouds, so as MODIS NDVI as well.

**Response #11**

We used MODIS NDVI images (23 scenes per year) from 2000 to 2007 (P1) and from 2010 to 2015 (P2), with 181 and 138 scenes in the two periods, respectively. During the whole study period, 53.64% of the observations have good quality while 46.36% were interpolated. For those pixels with less than 30 good-quality observations (4.79% in P1 and 9.64% in P2), we didn't apply the BFAST algorithm. For the remaining area, 61.67% (P1) and 58.24% (P2) of pixels had 12 (~50%) good-quality observations annually. We will further clarify it in the revised manuscript.

**Comment #12**

Eq 4, 5 and 6: some errors in explanation.

**Response #12**

We modified the statements on Section 2.4.2 Lines 257-268: " An ordinary least square residuals-based moving sum test (Zeileis 2005) was used to test whether breakpoints occurred in the trend or seasonal components. Then, test was conducted to determine the number and optimal position of the breaks using Bayesian Information Criteria and the minimum of the residual sum of squares. The trend and seasonal coefficients were then computed using a robust regression. A harmonic seasonality model (with three harmonic terms) was used to describe the seasonality of the satellite data (Eq. 6) (Verbesselt et al. 2010). For each piecewise linear ( $T_t$ ) from  $t_i^*$  to  $t_{i+1}^*$  where  $t_1^*, \ldots, t_p^*$  is the assumed break points which defines the p+1 segment,  $T_t$  can be expressed as follows:

$$T_t = \alpha_i + \beta_i t \ (i = 1, \dots, p)$$

(5)

where *i* is the index of the breaks, i=1, ..., *q*.  $\alpha_i$  and  $\beta_i$  are the intercept and slope of the fitted piecewise linear model.

For the  $t_1^{\#}, ..., t_m^{\#}$  seasonal break points,  $S_t$  is the harmonic model for  $t_j^{\#}$  to  $t_{j+1}^{\#}$ :

$$S_t = \sum_{k=1}^{K} \alpha_{j,k} \sin(\frac{2\pi kt}{f} + \delta_{j,k}) (j = 1, \dots, q)$$
(6)

where, j = 1, ..., q. *k* is the number of harmonic terms in the periodic model (default value = 3);  $\alpha_{j,k}$  is the amplitude; *f* is the frequency;  $\delta_{j,k}$  is the time phase. ".

**Comment #13**

More information is needed for the validation methods (2.5). E.g how many samples are there for each land use class for each year?

**Response #13**

We added the details and the number distribution of the validation sample set (Please see the **Table 2** (reproduced below) and the descriptions on **Section 2.5**, **Lines 311-315**: "Two sets of annual oil palm samples were used to validate the mapping results in Malaysia and Indonesia according to the sampling protocol of Gong et al. (2013). The independent annual sample set in Malaysia was from the previous studies (Cheng et al., 2019; Cheng et al., 2017). All pixel-based samples were randomly produced in equal-area hexagonal grid (95.98 km2 for each grid cell), therefore the distribution of the samples among different land cover types has minimum bias with the real land cover composition." And Lines 319-323: "The second annual Indonesia sample set was developed following the protocol of Cheng et al. (2017). This sample set contains 7663 samples in total (601 were oil palms and the rest were non-oil palm types) during 2010 to 2016 (see the blue points in Figure 3). The details of the number and spatial distribution of validation samples is presented in Figure 3 and Table 2. More information on the randomized sampling method could be referred to Cheng et al., 2017 and Cheng et al., 2019."

**Reference:**

Cheng, Y., Yu, L., Zhao, Y., Xu, Y., Hackman, K., Cracknell, A. P., and Gong, P.: Towards a global oil palm sample database: design and implications, Int. J. Remote Sens., 38, 4022-4032, 2017.

Cheng, Y., Yu, L., Xu, Y., Lu, H., Cracknell, A. P., Kanniah, K., and Gong, P.: Mapping oil palm plantation expansion in Malaysia over the past decade (2007–2016) using ALOS-1/2 PALSAR-1/2 data, Int. J. Remote Sens., 1-20, 2019

| Malaysia |          |                  |       |        |       |      | Indonesia |              |       |  |
|----------|----------|------------------|-------|--------|-------|------|-----------|--------------|-------|--|
|          | Oil palm | Other vegetation | Water | Others | Total |      | Oil palm  | Not oil palm | Total |  |
| 2007     | 371      | 2,335            | 68    | 74     | 2,848 | 2010 | 547       | 7066         | 7613  |  |
| 2008     | 398      | 2,334            | 71    | 76     | 2,879 | 2011 | 559       | 7063         | 7622  |  |
| 2009     | 418      | 2,335            | 71    | 76     | 2,900 | 2012 | 568       | 7068         | 7636  |  |
| 2010     | 433      | 2,335            | 71    | 76     | 2,915 | 2013 | 575       | 7078         | 7653  |  |
| 2015     | 505      | 2,336            | 75    | 76     | 2,992 | 2014 | 588       | 7072         | 7660  |  |
| 2016     | 505      | 2,334            | 71    | 73     | 2,983 | 2015 | 594       | 7073         | 7667  |  |
|          |          |                  |       |        |       | 2016 | 601       | 7066         | 7667  |  |
|          |          |                  |       |        |       | 2010 | 001       | 7000         | 7007  |  |

Table 2 The distribution of annual validation sample set for Malaysia and Indonesia (unit: pixel).

**Comment #14**

Line 726: Fig 3: are all the 2986 annual distribution of validation dataset is very uneven. There is no annual sample set in Sumatra Indonesia at all.

**Response #14**

We added a new annual validation sample set in Indonesia for the period from 2010 to 2016 to validate our datasets on **Section 2.5, Lines 319-323**. The datasets included 7667 samples in 2016 (601 samples were oil palm and the remaining were others – see above). The blue points in **Figure 3** (reproduced below) shows the spatial distribution of validation sample set in Indonesia. And **Table 4** (reproduced below) shows the validation results using the Indonesia annual sample.

**Figure 3** Spatial distribution of oil palm samples in the two validation datasets. The annual sample set contains 2986 (in 2016) samples in Malaysia which were interpreted for 2007, 2008, 2009, 2010, 2015 and 2016 and 7667 (in 2016) samples in Indonesia interpreted from 2010 to2016. These samples were

used to validate the annual maps developed from PALSAR/PALSAR-2 data. Of the annual sample set in Malaysia, oil palm samples consist of 16.92% (505) while the forest, water and others consist of 78.16%, 2.48% and 2.44%, respectively. The Indonesian annual sample set contains 601 (7.84%) oil palm samples and the rest (92.16%) were other types. The change sample set includes 370 oil palm samples which were converted in the interpolated period (2001-2006 and 2011-2014). This sample set, with change year labelled, is used to assess the change detection result in the gap years.

---

## Author Comment (AC2) · 31 Dec 2019

**Response to comments**

**Paper #:** essd-2019-137
**Title:** Annual oil palm plantation maps in Malaysia and Indonesia from 2001 to 2016
**Journal:** Earth System Science Data

**Reviewer #2:**

**General Comments:**

**Comment #1**

The manuscript is addressing the annual oil palm mapping in Malaysia and Indonesia from 2001 – 2016 by using PALSAR/PALSAR-2 imagery, and fill the PALSAR data gap (2011-2014) by using the MODIS data and the BFAST method. This study is well designed and the paper is very well written. But some parts should be further improved before its consideration for publication.

**Response #1**
We thank the reviewer for the comments and suggestions. Please see the detailed point-by-point responses below.

**Specific Comments:**

**Comment #1**

Effects of stand age. How the stand age could affect the identification of the oil palm plantation as well as the robustness of the BFAST approach? This study claims that the maps include young oil palm trees and smallholder oil palm plantations. What strategies have been considered to make sure the inclusion of young trees and smallholder plantations?

**Response #1**

We did a test to show the robustness of the algorithm at different age of oil palm plantation. Normally, the young oil palm (0-3 years old) was transplanted after the forest clearance, so the BFAST approach was applied to detect the conversion from forest to young oil palm at very young stage (the original planted age is referred as *young*). Here we manually moved forward the time-series NDVI after the break detected time to include older stand age and then re-applied the BFAST algorithm. For example, if the change year was detected at 2005, the subsequent 2006-2008 NDVI curves were replaced by 2007-2009 ones to show the effect of a one-year shift on the stand age (Here the age is referred as: *young*+1, if the 2008-2010 was used for two-year effect, the age is referred as *young*+2, etc.). Further, the break time detected by the new NDVI curves were compared with that of the original curves (differences of detected change time=break year$_{new}$ -break year$_{old}$). The differences among the different stand ages represented the effect of tree age and inform us about the algorithm's robustness. We applied the test for all the change pixels and **Figure S6** below shows the distribution of the differences between the new and original break time for all the results during 2000-2007. According to the result, the differences of detected change time were mostly concentrated on the values around zero (which mean there is no differences compared to the original detected change time) in all stand ages. In total, 79.69% (average result of the 7 stand ages) of the detected times show the agreement with the original result (76.73% of the detected years matched the original result while the rest were within one-year interval, **Figure S6**, reproduced below). This indicates the robustness of the algorithm under different stand ages and cloud conditions. With the increase of the stand age, the differences of the detected change time were increased (a 6.19% decrease of the agreement proportion presented if the tree is 6 years older than the other trees). However, the distribution pattern among the different stand ages is similar.

In the PALSAR mapping procedure, the training sample set used in the random forest classifier contains both young and mature oil palm samples (it could be identified by the canopy shape using very highresolution images from Google Earth in interpretation) therefore the outputs of the machine learning algorithm included young plantations. We will add these points in the revised manuscript.

The smallholder oil palm plantations were defined as: " oil palm smallholders is defined as 50 hectares or less of cultivated land producing palm oil controlled by smallholder farmers (the definition used by the RSPO) with an average of 2 ha (World Bank, 2010) " (Section 1, Lines 102-103), whereas our 1-ha mapping unit is able to depict some of the smallholder plantations between 1-50 ha.

**Figure S6** Effect of stand age. The values in x-axis is the difference between the detected change years using the replaced MODIS NDVI fragments (refer to older stand age) and the original NDVI curves (refer to young age). Negative values in x-axis refer to the detected change year using the older stand age is earlier than the original detected change year.

[Figure]

**Comment #2**

Effects of multiple data resolutions. Why does the resolution of 100m perform better to estimate oil palm planting area, not the 50m or other resolution? Is resolution of 100m sufficient to depict the smallholder details? Which resample technique did you use to resample 25-m PALSAR to 100-m? How did you integrate your 100-m oil palm maps with the 250-m land cover change maps?

**Response #2**

PALSAR data has a lot noise which may conceal the true land surface information. Filter analysis (Enhanced Frost, Enhanced Lee, Frost and Gamma filter) was compared with the resampling method at different resolution (25m, 50m, 100m, 250m, 500m, 1000m) in Cheng et al., (2018). The nearest neighborhood resampling at 100-m resolution showed the best mapping accuracy compared to the other filter methods and spatial resolution. Thus, we chose 100-m as the trade-off resolution of retaining the most land surface information as well as reducing noise.

Smallholders oil palm plantations are defined on an average of 2 ha and ranged up to 50 ha which is hold by family-based enterprises (Vermeulen and Goad, 2006; Lee et al., 2014 and World Bank 2010). Our results are able to capture part of the small oil palm plantations which are larger than 1 ha (100 m $\times$ 100 m).

We first identified the change area and "from-to" types in the 100-m land cover change maps. Then the MODIS product was resized to the same resolution as of 100-m land cover maps as described in **Section**

**2.4.1 Lines 222-224**: "All the MODIS images were projected from its original sinusoidal projection to a geographic grid with a WGS 1984 spheroid and resized to 100 m to match the resolution of the oil palm maps using the nearest neighbor resampling approach.)". Next, "We then sought the exact change year within the intervals in the next step (Section 2.4.2) using temporal NDVI files extracted from each change pixel. "as described in **Section 2.4.1, Lines 230-231**. Finally, "Change detection analysis was conducted in the change pixels derived from the last step to identify the exact change time within the two periods (2011-2014 and 2001-2006) based on the time-series MODIS NDVI from 2010 to 2015 and 2000 to 2007, respectively. " **(Section 2.4.1, Lines 238-240)**.

**Reference:**

Cheng, Y., Yu, L., Xu, Y., Lu, H., Cracknell, A. P., Kanniah, K., and Gong, P.: Mapping oil palm extent in Malaysia using ALOS-2 PALSAR-2 data, Int. J. Remote Sens., 39, 432-452, 2018.
Vermeulen, S., & Goad, N. (2006). Towards better practice in smallholder palm oil production. Iied.
Lee, J. S. H., Abood, S., Ghazoul, J., Barus, B., Obidzinski, K., & Koh, L. P. (2014). Environmental impacts of large-scale oil palm enterprises exceed that of smallholdings in Indonesia. Conservation letters, 7(1), 25-33.
World Bank. (2010) Improving the livelihoods of palm oil smallholders: the role of the private sector. International Finance Corporation, World Bank Group, Washington, DC, USA

**Comment #3**

How many types of land cover were got with the RF classification? Is the multi-class classification consistent with Table 1? Or the binary classification (oil palm; non-oil palm)?

**Response #3**

We got 4 land cover types (water, other vegetation, oil palm and others) from the RF classification. The result is consistent with multi-class classification. Here we presented the oil palm accuracy in the multi-class classification. As for the binary classification results, the average score of oil palm is 0.87/0.74 while the non-oil palm is 0.98/0.98 in Malaysia / Indonesia, respectively. For the newly added Indonesia validation sample set, we only have oil palm and non-oil palm types as described in **Section 2.5, Lines 321-322**: "This sample set contains 7663 samples in total (601 were oil palms and the rest were non-oil palm types) during 2010 to 2016."

**Comment #4**

You provided two version of oil palm datasets: one considers the oil palm expansion (unidirectional change) and the other one considers oil palm shrinkage (bi-directional change). Which version is more consistent with statistics? Which version is more accurate based on your validation samples? In Figure 5, the oil palm change in 2001-2007 is also unidirectional, thus the color of line might be blue, not green.

**Response #4**

The bi-directional version is more consistent with statistics. According to the validation sample, the unidirectional version is however more accurate (with an average 0.034 increase of *F*-score for each year). We changed the color in **Figure 5** (reproduced below) according to the suggestions.

**Figure 5:** Comparison of the annual oil palm plantation area among FAO and USDA statistics, MPOB records for Malaysia, BPS-Statistics and oil palm concessions from GFW for Indonesia and our mapping results in a) Malaysia, b) Indonesia and c) Malaysia and Indonesia from 2001 to 2016. The blue lines represent the gross gain (unidirectional expansion) while the green lines show the net changes of oil palm from 2007 to 2016. The shaded area within the two boundary lines are the uncertainty range of the oil palm area. The upper boundary lines represent the upper limit area of oil palm within the two periods (2011-2014 and 2001-2006), whereas the lower boundary lines are the lower limit according to our results. Note that during the gap between the two periods, no uncertainty could be derived, which does not mean that the uncertainty was small.

[Figure]

**Comment #5**

If there were more than one change time in 2011-2014 or 2001-2006, how did you allocate land cover types?

**Response #5**

We supposed more possibility of one-time change during such a short period other than the multi-time changes. For example, there is a long lead time (at least 2-4 years) between planting and productive harvest of oil palm and it is unlikely to do planting-cutting-replanting very often in such a short period, as described in **Section 2.4.1, Lines 231-232**: "Frequent changes such as two or three shifts during the gap years were assumed to be of low probability and thus not considered in this study." Therefore, we only consider the one-time change during the two time periods. We added the uncertainty caused by multiple changes in **Section 4.1, Lines 504-506**: "However, multiple changes may occur in the deforestation area when the logging activity is applied first and followed by the replantation of oil palm several years later.

---

## Author Comment (AC3) · 31 Dec 2019

**Response to comments**

**Paper #:** essd-2019-137
**Title:** Annual oil palm plantation maps in Malaysia and Indonesia from 2001 to 2016
**Journal:** Earth System Science Data

**Reviewer #1:**

**General Comments:**

**Comment #1**

The article presented the first annual oil palm plantation maps in Malaysia and Indonesia and demonstrate the accuracy of the maps through various comparisons with existing statistic dataset and regional maps. It's an interesting paper that exhibits the efficiency of fusing optical and radar data in over coming data gaps to produce consistent annual maps. However, there are quite a few details in the abstract and introduction session that need to be checked. Some statement are lacking adequate references. More detail needs to be given on the methods, especially validation approach. Some of the conclusions in the discussion section need to be backed up, either by reference or by results. I'm not very convinced by the results due to limited information was given to the independence validation approach.

**Response #1**

We thank the reviewer for the comments and suggestions. Please see the detailed point-by-point responses below.

**Specific Comments:**

**Comment #1**

Abstract/Introduction:
Line 12: The land convention to oil palm plantations not always lead to deforestation.

**Response #1**

Oil palm conversion takes places not only in forest but also agroforests, agricultural fallows, bare lands and etc. So we changed the original sentence to "The land conversion to oil palm plantations poses risks to deforestation (50% of the oil palm was taken from forest during 1990-2005, Koh and Wilcove, 2008), loss of biodiversity, and greenhouse gas emission over the past decades." **(Abstract, Lines 12-14)**.

**Comment #2**

Line 26: Current discussion is not strong enough to support the conclusion that the higher trend in this study is due to the inclusive of smallholder farmers. (more comments in the Results part, section 3.3)

**Response #2**

We totally agree with this. The inclusive of smallholder farmers is one of the potential reasons of the higher trend in this study. We rewrote the conclusions and excluded it in the abstract **(Abstract, Lines 26-28)**: "The higher trends from our dataset are consistent with those from the national inventories with limited annual average difference in Malaysia (0.2 M ha) and Indonesia (-0.17 M ha)." And we also discussed more possible reasons in the Result and discussion part (Please see the reply to comment#19).

**Comment #3**

Line 36: Corley, 2009- any more recent ref to support the expected growing rate from 2003?

**Response #3**

We updated the growing rate of oil palm fruit production in Malaysia and Indonesia to 2017 according to FAO statistics and added a new reference projecting a considerable expansion of oil palm cultivation worldwide in the future in **Section 1, Lines 35-37** :"According to the Food and Agriculture Organization (FAO), Malaysia and Indonesia account for 81.90% of the global oil palm fruit production in 2017, an increase by 179.72% from 2000 to 2017 (see http://faostat.fao.org) that is projected to continue in the future (Murphy, 2014). "

**Reference:**

*Murphy, D. J. (2014). The future of oil palm as a major global crop: opportunities and challenges. J Oil Palm Res, 26(1), 1-24.*

**Comment #4**

Line 38:"forest cover dropped from 76% to 9% since 1990 in Malaysia and Indonesia". Please double check these numbers, and cross reference with other sources.

**Response #4**

Sorry for the mistake. The peat swamp forest dropped from 76% to **2**9% since 1990 in Malaysia and Indonesia according to the reference. We also added references to show the deforestation caused by oil palm expansion on **Section 1 Lines 38-41**:" In Malaysia and Indonesia, more than 50% of the oil palm plantation was converted from forest during 1990-2005 (Koh and Wilcove, 2008) and industrial plantations dominated by oil palm (72.5% of all plantations) caused a 60% decrease of peatland forest from 2007 to 2015 (Miettinen et al., 2016)."

**Comment #5**

Line 43: There are quite a few existing dataset/report that are providing continuous information about the expansion of oil palm in Indonesia and Malaysia. E.g. https://theicct.org/sites/default/files/publications/ICCT_palm- expansion_Feb2012.pdf

**Response #5**

We thank the reviewer for this information. We added the references of the continuous mapping of oil palm on **Section 1, Lines 50-52**: "The continuous mapping of oil palm on peatland in 1990, 2000, 2007 and 2010 described the dynamic change of oil palm on peat during the past 30 years (Miettinen et al., 2012)." Here we also modified the text from continuous to annual mapping in **Section 1 Line 45**: " However, annual information on the expansion of oil palm plantations is poorly documented in Malaysia and Indonesia."

**Comment #6**

Line 59: There are quite a lot of Machine learning or Deep Learning based methods for automatic identification of oil palms.

**Response #6**

  We added the recent deep learning based automatic identification references here as suggested on **Section 1, Line 63-66**: "3) interpretation methods from manual to semi- and fully automatic identification (Baklanov et al., 2018; Cheng et al., 2019; Li et al., 2017a; Mubin et al., 2019; Ordway et al., 2019), 4) products going from oil palm land cover maps to more detailed datasets on plantation structure, e.g. tree counting (Li et al., 2019; Cheang et al., 2017)."

**Reference:**

*Baklanov, A., Khachay, M., and Pasynkov, M.: Application of fully convolutional neural networks to mapping industrial oil palm plantations, International Conference on Analysis of Images, Social Networks and Texts, 2018, 155-167,*

*Cheang, E. K., Cheang, T. K., and Tay, Y. H. J. a. p. a.: Using convolutional neural networks to count palm trees in satellite images, 2017.*

*Li, W., Fu, H., Yu, L., and Cracknell, A. J. R. S.: Deep learning based oil palm tree detection and counting for high-resolution remote sensing images, Remote Sensing, 9, 22, 2017a.*

*Mubin, N. A., Nadarajoo, E., Shafri, H. Z. M., and Hamedianfar, A.: Young and mature oil palm tree detection and counting using convolutional neural network deep learning method, Int. J. Remote Sens., 40, 7500-7515, 10.1080/01431161.2019.1569282, 2019.*
* * *
**Comment #7**

**Methods:**
Any co-registration issue between MODIS and ALOS/ALOS2?

**Response #7**

We've checked there is no co-registration issues. And other analysis was also directly conducted in MODIS and PALSAR data in previous researches (Qin et al., 2017 and Zhang et al., 2019). We will further clarify this point in the revised manuscript.

**Reference:**

*Zhang, Y., Ling, F., Foody, G. M., Ge, Y., Boyd, D. S., Li, X., Du, Y., and Atkinson, P. M.: Mapping annual forest cover by fusing PALSAR/PALSAR-2 and MODIS NDVI during 2007–2016, Remote Sens. Environ., 224, 74-91, https://doi.org/10.1016/j.rse.2019.01.038, 2019.*

*Qin, Y., Xiao, X., Dong, J., Zhou, Y., Wang, J., Doughty, R. B., Chen, Y., Zou, Z., and Moore, B.: Annual dynamics of forest areas in South America during 2007–2010 at 50-m spatial resolution, Remote Sens. Environ., 201, 73-87, https://doi.org/10.1016/j.rse.2017.09.005, 2017.*
* * *
**Comment #8**

Line 149: Any other prove that no calibration is needed between ALOS and ALOS2 in Indonesia and Malaysia? The study site for the two referenced papers are not for these two coun-tries specifically (Thus with different incident angel, weather condition, etc).

**Response #8**

We randomly generated 250,000 points using ArcGIS 10.3 in our study area and compared the HH/HV values of these points during the 6 years following Qin et al (2016) and Cheng et al (2019)'s practice (**Figure S2**, reproduced below). According to the histogram, the backscattering value of PALSAR/PARSAR-2 are relatively stable in the study period. The reference which presented the stability of annual PALSAR/PALSAR-2 HH and HV values in Malaysia was also added (Cheng et al., 2019). Meanwhile, the HH and HV values for oil palm and forest is also shown in **Figure S3** (reproduced below) and indicate the separability between the two land cover types for both PALSAR/PALSAR-2 data. We will add these points in the revised manuscript. We produced the classification map using the training samples from each corresponding year, the influence of calibration differences between PALSAR/PALSAR-2 data will not influence the mapping results.

**Reference:**

*Cheng, Y., Yu, L., Xu, Y., Lu, H., Cracknell, A. P., Kanniah, K., and Gong, P.: Mapping oil palm plantation expansion in Malaysia over the past decade (2007–2016) using ALOS-1/2 PALSAR-1/2 data, Int. J. Remote Sens., 1-20, 2019*

*Qin, Y., Xiao, X., Dong, J., Zhou, Y., Wang, J., Doughty, R. B., Chen, Y., Zou, Z., and Moore, B.: Annual dynamics of forest areas in South America during 2007–2010 at 50-m spatial resolution, Remote Sens. Environ., 201, 73-87, https://doi.org/10.1016/j.rse.2017.09.005, 2017.*

**Figure S2** Density distribution of PALSAR/PALSAR-2 (a) HH (dB) and (b) HV (dB) in study area for 2007, 2008, 2009, 2010, 2015 and 2016 based on 250000 randomly generated points. The mean and standard deviation (std) value for the six years were given (mean: -7.44~-6.98 of HH and -13.47~-13.01

of HV; std: 2.52~2.90 of HH and 3.05~3.76 of HV). According to the result, the backscatter signals are relatively stable for the given period (2007–2010 and 2015–2016).

[Figure]

**Figure S3** Comparison between PALSAR/PALSAR-2 (a) HH (dB) and (b) HV (dB) for forest and oil palm based on the training points. The HV (dB) for the forest and oil palm samples are differentiable during the given period (2007–2010 and 2015–2016).

[Figure]

**Comment #9**

Line 108: How dose 98.91% been calculated?

**Response #9**

We updated the number (96%) according to the reference (Petrenko et al., 2016) on **Section 2. 1, Line 113-115**: "Thus, we chose as a study area the whole Malaysia, Sumatra and Kalimantan in Indonesia, encompassing 96% of the total oil palm production in Indonesia (Petrenko et al., 2016)."

**Reference:**

Petrenko, C., et al. (2016). "Ecological impacts of palm oil expansion in Indonesia." J Washington : International Council on Clean Transportation.

**Comment #10**

Why the NDIV information from MODIS is not used as input to the RF model for classification?

**Response #10**

The use of coarse resolution MODIS information in RF may negate the benefits of our classification based on higher spatial resolution PALSAR data, keeping in mind that the change detection results during the gap years is based on the results from that classification. Second, we also found that the spectral information used to derive NDVI is quite similar between a tropical forest and a mature oil palm plantation, which induces confusion in the classification (Razak., 2018). Some studies used the fusion method (such as super-resolution mapping) to fusing coarser resolution MODIS with higher resolution PALSAR data, but these algorithms require large computational cost and were always applied to small scenes. For these two reasons, we didn't include the MODIS NDVI in the RF model. We will further add these points in the revised manuscript.

**Reference:**

*Razak, J. A. B. A., Shariff, A. R. B. M., Ahmad, N. B., & Ibrahim Sameen, M. (2018). Mapping rubber trees based on phenological analysis of Landsat time series data-sets. Geocarto international, 33(6), 627-650.*

**Comment #11**

Line 213: How many MODIS time series are used exactly? How many are actual data and how many are interpolated? As the author explained, Indonesia and Malaysia are heavily affected by clouds, so as MODIS NDVI as well.

**Response #11**

We used MODIS NDVI images (23 scenes per year) from 2000 to 2007 (P1) and from 2010 to 2015 (P2), with 181 and 138 scenes in the two periods, respectively. During the whole study period, 53.64% of the observations have good quality while 46.36% were interpolated. For those pixels with less than 30 good-quality observations (4.79% in P1 and 9.64% in P2), we didn't apply the BFAST algorithm. For the remaining area, 61.67% (P1) and 58.24% (P2) of pixels had 12 (~50%) good-quality observations annually. We will further clarify it in the revised manuscript.

**Comment #12**

Eq 4, 5 and 6: some errors in explanation.

**Response #12**

We modified the statements on **Section 2.4.2 Lines 257-268**: " An ordinary least square residuals-based moving sum test (Zeileis 2005) was used to test whether breakpoints occurred in the trend or seasonal components. Then, test was conducted to determine the number and optimal position of the breaks using Bayesian Information Criteria and the minimum of the residual sum of squares. The trend and seasonal coefficients were then computed using a robust regression. A harmonic seasonality model (with three harmonic terms) was used to describe the seasonality of the satellite data (Eq. 6) (Verbesselt et al. 2010). For each piecewise linear $(T_t)$ from $t_i^*$ to $t_{i+1}^*$ where $t_1^*, \ldots, t_p^*$ is the assumed break points which defines the p+1 segment, $T_t$ can be expressed as follows:

$$T_t = \alpha_i + \beta_i t \ (i = 1, \ldots, p) \tag{5}$$

where $i$ is the index of the breaks, i=1, ..., q. $\alpha_i$ and $\beta_i$ are the intercept and slope of the fitted piecewise linear model.

For the $t_1^\#, \ldots, t_m^\#$ seasonal break points, $S_t$ is the harmonic model for $t_j^\#$ to $t_{j+1}^\#$:

$$S_t = \sum_{k=1}^{K} \alpha_{j,k} \sin\left(\frac{2\pi k t}{f} + \delta_{j,k}\right) (j = 1, \ldots, q) \tag{6}$$

where, $j = 1, \ldots q$. $k$ is the number of harmonic terms in the periodic model (default value = 3); $\alpha_{j,k}$ is the amplitude; $f$ is the frequency; $\delta_{j,k}$ is the time phase. ".

**Comment #13**

More information is needed for the validation methods (2.5). E.g how many samples are there for each land use class for each year?

**Response #13**

We added the details and the number distribution of the validation sample set (Please see the **Table 2** (reproduced below) and the descriptions on **Section 2.5, Lines 311-315**: " Two sets of annual oil palm samples were used to validate the mapping results in Malaysia and Indonesia according to the sampling protocol of Gong et al. (2013). The independent annual sample set in Malaysia was from the previous studies (Cheng et al., 2019; Cheng et al., 2017). All pixel-based samples were randomly produced in equal-area hexagonal grid (95.98 km$^2$ for each grid cell), therefore the distribution of the samples among different land cover types has minimum bias with the real land cover composition." And **Lines 319-323**: " The second annual Indonesia sample set was developed following the protocol of Cheng et al. (2017). This sample set contains 7663 samples in total (601 were oil palms and the rest were non-oil palm types) during 2010 to 2016 (see the blue points in Figure 3). The details of the number and spatial distribution of validation samples is presented in Figure 3 and Table 2. More information on the randomized sampling method could be referred to Cheng et al., 2017 and Cheng et al., 2019."

**Reference:**

*Cheng, Y., Yu, L., Zhao, Y., Xu, Y., Hackman, K., Cracknell, A. P., and Gong, P.: Towards a global oil palm sample database: design and implications, Int. J. Remote Sens., 38, 4022-4032, 2017.*
*Cheng, Y., Yu, L., Xu, Y., Lu, H., Cracknell, A. P., Kanniah, K., and Gong, P.: Mapping oil palm plantation expansion in Malaysia over the past decade (2007–2016) using ALOS-1/2 PALSAR-1/2 data, Int. J. Remote Sens., 1-20, 2019*

**Table 2** The distribution of annual validation sample set for Malaysia and Indonesia (unit: pixel).

| | Malaysia | | | | | | Indonesia | | |
|---|---|---|---|---|---|---|---|---|---|
| | Oil palm | Other vegetation | Water | Others | Total | | Oil palm | Not oil palm | Total |
| 2007 | 371 | 2,335 | 68 | 74 | 2,848 | 2010 | 547 | 7066 | 7613 |
| 2008 | 398 | 2,334 | 71 | 76 | 2,879 | 2011 | 559 | 7063 | 7622 |
| 2009 | 418 | 2,335 | 71 | 76 | 2,900 | 2012 | 568 | 7068 | 7636 |
| 2010 | 433 | 2,335 | 71 | 76 | 2,915 | 2013 | 575 | 7078 | 7653 |
| 2015 | 505 | 2,336 | 75 | 76 | 2,992 | 2014 | 588 | 7072 | 7660 |
| 2016 | 505 | 2,334 | 71 | 73 | 2,983 | 2015 | 594 | 7073 | 7667 |
| | | | | | | 2016 | 601 | 7066 | 7667 |

**Comment #14**

Line 726: Fig 3: are all the 2986 annual distribution of validation dataset is very uneven. There is no annual sample set in Sumatra Indonesia at all.

**Response #14**

We added a new annual validation sample set in Indonesia for the period from 2010 to 2016 to validate our datasets on **Section 2.5, Lines 319-323**. The datasets included 7667 samples in 2016 (601 samples were oil palm and the remaining were others – see above). The blue points in **Figure 3** (reproduced below) shows the spatial distribution of validation sample set in Indonesia. And **Table 4** (reproduced below) shows the validation results using the Indonesia annual sample.

**Figure 3** Spatial distribution of oil palm samples in the two validation datasets. The annual sample set contains 2986 (in 2016) samples in Malaysia which were interpreted for 2007, 2008, 2009, 2010, 2015 and 2016 and 7667 (in 2016) samples in Indonesia interpreted from 2010 to2016. These samples were

used to validate the annual maps developed from PALSAR/PALSAR-2 data. Of the annual sample set in Malaysia, oil palm samples consist of 16.92% (505) while the forest, water and others consist of 78.16%, 2.48% and 2.44%, respectively. The Indonesian annual sample set contains 601 (7.84%) oil palm samples and the rest (92.16%) were other types. The change sample set includes 370 oil palm samples which were converted in the interpolated period (2001-2006 and 2011-2014). This sample set, with change year labelled, is used to assess the change detection result in the gap years.

[Figure]

**Table 4** The oil palm accuracy in Indonesia from 2010-2016. UA: User's Accuracy; PA: Producer's Accuracy

| Year | Our results | | |
|---|---|---|---|
| | *F*-score | UA (%) | PA (%) |
| 2010 | 0.75 | 69.47 | 74.95 |
| 2011 | 0.75 | 70.38 | 74.83 |
| 2012 | 0.75 | 71.48 | 75.05 |
| 2013 | 0.75 | 72.39 | 74.79 |
| 2014 | 0.74 | 72.58 | 74.28 |
| 2015 | 0.72 | 68.46 | 71.83 |
| 2016 | 0.72 | 69.97 | 72.33 |

**Comment #15**

Line 300: How does the total number of validation points (5000) been decided? What's the ratio of the validation points to the total pixel been detected as change?

**Response #15**

We randomly generated 5000 samples in the change areas (which should all be changed area according to our results). However, as the lack of continuous high-resolution images from Google Earth and cloud-free Landsat time series, 370 samples were manually interpreted with actual change years and used as the change sample set. In total there are 370 changed oil palm samples in 1476 (25.07%) oil palm

samples and 10500 total samples, whereas the ratio is 25.07% and 3.52%, respectively. We will further clarify this point in the revised manuscript.

**Comment #16**

**Results:**
Paragraph 1 and 2: There is no other information/ref/map/graph/table provided to support many of the conclusions in these two paragraphs. Some of the sentences read like discussion rather than results.

**Response #16**

We added a SI figure (**Figure S5**, reproduced below) of the oil palm distribution according to elevation and slope topography and rewrote the unclear sentences in these two paragraphs: "In the study area, most oil palm plantations are located on lowland areas (elevation <250 m, slope <2.5 degree), and few are distributed in gently undulating hills (elevation >500 m, slope >5 degree) (Figure S5). The newly developed oil palm has similar elevation and slope distribution compared to the 2007 ones (slope: 1.97° in 2007/1.99° in 2016; elevation 228.98 m in 2007/230.10 m in 2016)" (**Section 3.1, Lines 334-337**) and "In Indonesia, rapid expansion first occurred in Sumatra and was then surpassed by Kalimantan (Gunarso, 2013; Petrenko et al., 2016). This can also be observed in our maps where more changes happened in earlier years in Sumatra (lighter colors in Figure 4 of the revised manuscript) and later in Kalimantan (darker colors)." (**Section 3.1, Lines 341-343**).

**Figure S5:** Frequency histograms of elevation and slope for oil palm distribution in 2007 and 2016 over the study area. According to the results, the oil palm is mainly distributed on the lowland areas (elevation <250 m, slope <2.5 degree).

[Figure]

**Reference:**

*Gunarso, P., Hartoyo, M., Agus, F. & Killeen, T.: Oil palm and land use change in Indonesia, Malaysia and Papua New Guinea, 2013.*
*Petrenko, C., et al. (2016). "Ecological impacts of palm oil expansion in Indonesia." J Washington : International Council on Clean Transportation.*

**Comment #17**

Section 3.3: Have you compared your results from Global forest watch, oil palm concession dataset 2014?

**Response #17**

Thank you for this suggestion. We added the comparison the spatial distribution with PALSAR data and area with oil palm concession from Global forest watch on **Section 3.3 Lines 451-460** and **Figure 9** (reproduced below): " The oil palm concession area for Indonesia and Malaysia (Sarawak) for 2014 from global forest watch (www.globalforestwatch.org) is also used in the comparison. This dataset indicated the boundaries of areas allocated by government to companies for oil palm plantation. The oil palm concession area in Indonesia and Malaysia (Sarawak) for 2014 is 12.98 M ha, which is slightly higher (8.7%) than our mapping results (11.85 M ha). However, since the concession data was compiled from various countries and sources (such as governments and other organizations) with different quality, some location of the existing concessions may be inaccurate (**Figure 9(a)**) or omitted. Another possible reason for the differences is the inclusion of very small oil palm plantations in our dataset of less than 50 ha, while most of the oil palm concessions (81.71%) were larger than 1000 ha."

**Reference:**

*Slette, J. P., and I. E. Wiyono. 2011. Oilseeds and products update 2011. USDA Foreign Agricultural Service, Washington, D.C., USA. [online] URL:*
*http://www.usdaindonesia.org/public/uploaded/Oilseeds%20and%20Products%20Update_Jakarta_Indonesia_1-28-2011.pdf*

**Figure 9** Comparison with oil palm concession from Global forest watch (GFW) for year 2014. The PALSAR-2 images were composited in RGB format (HH, HV, HV).

[Figure]

**Comment #18**

Section 3.3: There lacks adequate reference to support the linkage between oil palm expansion, price fluctuation.

**Response #18**

It is difficult to conclude the relationship between oil palm expansion and the price fluctuations since the plantation area is affected by multiple price-related factors such as land rent and production tax. We modified the texts on **Section 3.3, Lines 425-431**: "During the study period, the oil palm export price (total export value/export amount, data source: FAOSTAT) rapidly increased from 402.67 dollars/t in 2006 to the peak (1080.72 dollars/t) in 2011 (Figure S9, Figure 8 in the old version) but subsequently fell. The crop price is closely related to demand and may further impact the oil palm market and

production (Turner et al., 2011). However, although there is a ~10-20% slowdown of the conversion rate, oil palm plantation area continuously increased after 2011. The land conversion to oil palm may also be affected by multiple factors such as agricultural rent, wages and market-mediated effects (such as tax) (Furumo and Aide, 2017; Taheripour et al., 2019), and the relationship between oil palm expansion and price fluctuation still requires further exploration." and put the price figure to supplementary (**Figure S9**).

**Reference:**

*Furumo, P. R., and Aide, T. M. J. E. R. L.: Characterizing commercial oil palm expansion in Latin America: land use change and trade, 12, 024008, 2017.*
*Taheripour, F., Hertel, T. W., and Ramankutty, N.: Market-mediated responses confound policies to limit deforestation from oil palm expansion in Malaysia and Indonesia, Proceedings of the National Academy of Sciences, 116, 19193, 10.1073/pnas.1903476116, 2019.*
*Turner, E. C., Snaddon, J. L., Ewers, R. M., Fayle, T. M., and Foster, W. A. J. E. i. o. b.: The impact of oil palm expansion on environmental change: putting conservation research in context, 10, 20263, 2011*

**Comment #19**

Section 3.3: There are potentially more reasons to explain the higher estimated oil palm area in this study compared to existing dataset. More evidence is needed to exclude other reasons and draw the conclusion to smallholders' oil palm plantation. Especially the minimum mapping unit in this paper is 1ha.

**Response #19**

We added more discussion about the higher estimation in **Section 3.3**: "The higher estimation may be induced by the confusion in other woody plantations such as coconuts and pulp. Although there is high separability between rubber, wattles and palms in PALSAR data (Miettinen and Liew, 2011), the coconuts which belongs to palm trees and have a fan-like shape showed less differences with oil palm compared to other plantations" (**Section 3.3**, **Lines 419-422**), " We should also note that the uni-directional version would have a higher estimation of oil palm plantation area since the assumption of one-way growth" (**Section 3.3**, **Lines 426-427**), " The oil palm concession area in Indonesia and Malaysia (Sarawak) for 2014 is 12.98 M ha, which is 8.7% higher than our mapping results (11.85 M ha). However, since the concession data was compiled from various countries and sources (such as government and other organizations) with different quality, some location of the existing concessions can be inaccurate (Figure 9(a)) or may be omitted (Figure 9(b)) comparing the concessions and our mapping results with PALSAR-2 data. Many concessions are not fully developed and the number reached more than 11 M ha (more than half) in 2010. Another possible reason for the differences may be the inclusive of oil palm plantations less than 50 ha in our results, while most of the oil palm concessions (81.71%) were larger than 1000 ha." (**Section 3.3**, **Lines 451-460**). And we also explained the uncertainty of the datasets in discussion part, "…but confusion may occur in some impervious area and plantations of other species such as coconuts. As a result, the accuracy of the change detection in the second step was also influenced by the oil palm maps generated from PALSAR/PALSAR-2 data in the first stage... inaccurate inputs in some pixels may lead to cumulative errors in the change detection during the PALSAR data gap years, particularly in Indonesia. " (**Section 4.1**, **Lines 482-487**), " … the use of moderate resolution MODIS data at 250 m may cause the loss of spatial information and false identification of the change times. … In addition to the satellite data, the change detection algorithm may also bring uncertainties. Because the accuracy of the detected change time by BFAST within a time series is influenced by the signal-to-noise ratio (Verbesselt et al., 2010b), cloud contamination and poor data quality in some regions from MODIS reduced the amount of valid information. And the bias may also be found in the gap years when no breakpoint could be found using BFAST algorithm and the errors were accumulated to years when switching to MODIS before and after PALSAR. " (**Section 4.1**, **Lines 490-500**)

As for the concern of mapping units and smallholders, on average, each farming household manages about 2 ha of land (ranged up to 50 ha), compared with private companies that manage about 4,000 ha (Daemeter Consulting 2015, Vermeulen and Goad, 2006; Lee et al., 2014). Compared to the existing industrial oil palm plantation datasets (81.71% are larger than 1000 ha in GFW oil palm concession), our datasets included oil palm plantation larger than 1 ha which contains some of the small-scale family-based enterprises. But the spatial resolution still limits the detection of smallholder less than 1 ha. We believe it is reasonable to attribute part of our higher estimated oil palm area to smallholder land but not all the differences. Therefore, we also modified the statements in the manuscript: "Our higher estimation of oil palm plantation area is possibly because some of the smallholders oil palm plantations (1-50 ha in size) is captured in our results whereas only industrial plantations were visually interpreted in Gaveau's results. Misclassification (commission errors) in our results may however also contribute to our estimation being higher. " (**Section 4.1**, **Lines 447-450**)

**Reference:**

*Daemeter Consulting (2015): Indonesian Oil Palm Smallholder Farmers: A Typology of Organizational Models, Needs, and Investment Opportunities. Daemeter Consulting, Bogor, Indonesia*

*Vermeulen, S., & Goad, N. (2006). Towards better practice in smallholder palm oil production. Iied.*

*Lee, J. S. H., Abood, S., Ghazoul, J., Barus, B., Obidzinski, K., & Koh, L. P. (2014). Environmental impacts of large-scale oil palm enterprises exceed that of smallholdings in Indonesia. Conservation letters, 7(1), 25-33.*

**Comment #20**

Line 435: what does 'limited bands in ALOS/ALOS 2 mean?

**Response #20**

Here we mean there is two bands (HH HV) in the original data. We deleted the inaccurate description.

---

## Author Response (AR2)

**Response to comments**

**Paper #:** essd-2019-137
**Title:** Annual oil palm plantation maps in Malaysia and Indonesia from 2001 to 2016
**Journal:** Earth System Science Data

**Reviewer:**

**General Comments:**

**Comment #1**

Overall a very good product, well-documented and with good validation and uncertainty analyses. A few remaining concerns:

**Response #1**

We thank the reviewer for the comments and suggestions. Please see the detailed point-by-point responses below.

**Specific Comments:**

**Comment #1**

Lines 107, 108 - "is the most biologically diverse terrestrial ecosystem on Earth": unfortunately, many regions of the planet claim this distinction: temperate rainforests, high-elevation regions with multiple water/soil/exposure gradients, etc. Unless authors have a specific citation for this statement, or want to get into 'greatest biodiversity' arguments, I suggest they write 'is one of the most'

**Response #1**

We modified the statement to as suggested, "Insular South-East Asia was originally occupied by evergreen moist tropical forest, which is one of the most biologically diverse terrestrial ecosystem on Earth." **(Abstract, Lines 107-108)**.

**Comment #2**

Line 180 - "used a random forest (RF) classifier, a robust, stable and efficient machine": many other researchers apply RF techniques (often referenced in ESSD) but they usually cite a reference, particularly if they claim 'robust' and 'efficient'. Do these author choose to buttress this claim with citation(s)? If not, perhaps more caution in the statement?

**Response #2**

We totally agree with this. We rewrote the sentences to :"Thereafter, we used a random forest (RF) classifier in the image classification step."**(Section 2.3.2, Lines 181)**

**Comment #3**

Line 203 - "trees will be destroyed and transplanted": Unless I mis-understand oil palm biology, a destroyed tree can not be subsequently transplanted? Older non-productive trees destroyed and plantation renewed by transplantation of new younger plants?

**Response #3**

Sorry for the misleading statement. Here we mean that the oil palm tree will be cleared at the age about 25 years and then another young oil palm tree will be transplanted here. We changed the text here and add the reference in **Section 2.3.3, Lines 203-204** :"Oil palm has a long-life cycle of 25 to 30 years. After that, the trees will be cleared and replaced because of a decrease in palm oil yield (Röll et al, 2015)".

**Reference:**

*Röll, A., Niu, F., Meijide, A., Hardanto, A., Knohl, A., & Hölscher, D. (2015). Transpiration in an oil palm landscape: effects of palm age. Biogeosciences, 12(19), 5619-5633.*

**Comment #4**

Line 226-228 - "For those pixels with less than 30 good-quality observations (4.79% in P1 and 9.64% in P2), we didn't apply the BFAST algorithm. For the remaining area, 61.67% (P1) and 58.24% (P2) of pixels had 12 (~50%) good-quality observations annually." I think a reviewer also asked about these numbers. The latter number of "12 (~50%) good-quality observations annually" refers to the 23 MODIS NDVI image per year (line 220)? But where does the '30' come from in "less than 30 good-quality"? Not from the 181 (P1) or 138 (P2) total scene because the percentage don't match? Please clarify.

**Response #4**

Here we mean that 4.79% and 9.64% pixels in P1 and P2 has less than 30 observations and for these pixels we didn't apply the BFAST algorithm because the number of observations is too small. The percentage here is the pixels in the total change area, while 50% is the percentage of 12 in 23 MODIS NDVI observations per year exactly. We modified the sentences to make it clear:" For those pixels with less than 30 good-quality observations (4.79% in P1 and 9.64% in P2 of the total change area), we didn't apply the BFAST algorithm. For the remaining area, 61.67% (P1) and 58.24% (P2) of pixels had 12 (~50% in 23) good-quality observations annually. " **Section 2.4.1 Lines 227-229.**

**Comment #5**

Lines 248-250 - "Considering the consistency in capturing significant changes (e.g., logging and replanting), predefined single conversion and computation volume, we used one of the commonly used change detection algorithms, BFAST, to capture the oil palm conversion time within the two periods (2011-2014 and 2001-2006)." Yes, but this sentence merely repeats what the authors wrote at the start of this paragraph. Authors should consider if they need this sentence or if in fact they intended to make a different point?

**Response #5**

We deleted the repeated sentences as suggested (**Section 2.4.2 Lines 249)**.

**Comment #6**

Line 280 - "break time detected from MODIS NDVI showed the same conversion year compared with the microwave satellite images." Good important statement but readers should get a figure, an R2 value, something to validate this statement? As one reviewer requested, we need better information to certify this statement in order to build confidence in the remaining (temporal) assertions. Later (around line 295) a reader finds reference to "consistency of change time detection from different breakpoint test approaches" referring to figure S(upplement)7, but - to the extent that correspondence of the MODIS-determined time changes and the SAR-determined time changes serves as a fundamental assumption of the paper and as crucial link between two primary data sources, a reader needs clearer quantitative proof here? I see this request as entirely consistent with comments from the reviewer who questioned whether authors had demonstrated sufficient basis for their conclusions. Later (line 352) a reader learns that MODIS NDVI failed to capture ~ 10% of land use changes? Where did this number come from?

**Response #6**

Thanks for your suggestion. We adopted BFAST in P1 (2000-2007) and P2 (2010-2015) where no PALSAR images are available. And no changes were detected in MODIS NDVI in ~10% of the total PALSAR-determined changed area based on 2010 and 2015 PALSAR images. This may due to the coarse spatial resolution and cloud cover. However, we cannot directly compare the two MODIS and PALSAR result in the two study period (P1 and P2) without PALSAR images. Therefore, we used a change sample set which is visually interpreted based on high-resolution images from Google Earth and 75.74% of the samples showed the same detected and the actual change time (**Section 3.2, the last paragraph and Figure 7**).

However, it is also important to check the consistency between the MODIS- and the SAR-determined time changes as suggested. So we visually interpreted an extra change sample set (100 points) based on the PALSAR images (2007-2010) and applied the BFAST algorithms using the MODIS NDVI. By comparison, there is 86% (62% matched the same change year while 24% where within a 1-year interval) agreement between the PALSAR and MODIS based change time. We added this point in **Section 2.4.2 Line 275-280**.
* * *
**Comment #7**

Lines 317-320: apparently authors had access to 3000 high-quality high-resolution GE images for Malaysia for the time period 2001 to 2016 (15 years) (later, more than 7000 for Indonesia). Over many regions of the globe, high-resolution GE images remain sparse, inconsistent and unreliable, particularly before 2010 or even - for some regions - 2015. Please can authors assure readers about this apparently positive availability of high-quality GE images? Figure 3 and Table 2 show only locations and cumulative numbers of images. All those locations and land-use types relied on high-res GE images? Lines 329-330 "limited samples (370, 25.07% of the total 1476 oil palm samples) with continuous high-resolution images from Google Earth" refers to exactly this problem of spare inconsistent GE images. Perhaps I have missed something about processing steps, but if so authors need to clarify for me and for other readers.

**Response #7**

Exactly, the high-resolution GE images is sparse and inconsistent. In most of the cases, we only accessed the high-resolution images through GE in some time points (e.g., we found high-resolution GE images in 2008, 2014 and the year after 2017). And if change happened during the time, the actual change year is not determined. Therefore, only limited samples with continuous high-resolution images from Google Earth was found in the change validation sample set as mentioned.

As for the annual sample set in Table 2, this is interpreted by both high-resolution GE image and PALSAR **(Section 2.5, Lines 318-320)**. "
[revised manuscript text omitted]

870